# Endoglin as a BMP9 co-receptor in vascular endothelial cells: prodomain displacement and TGFBRII recruitment

Jingxu Guo [1], Karolina Kostrzyńska [1], Ioannis Kamzolas [2,3], Xudong Yang[1], Midory Thorikay[4], Eckart De Bie [1], Rowena J. Jones[1], Adam Brownstein[5], Lu Long[1], Christopher J. Rhodes [6], Allan Lawrie [6], Martin R. Wilkins [6], Esmee Groeneveld[4], Zhen Tong[1], Marie-José Goumans [4], Evangelia Petsalaki [3], Jason Hong [5], Mark R. Toshner[1], Antonio Vidal-Puig[1,2,7], Helen M. Arthur[8] & Wei Li [1]✉

Endoglin (ENG) is a single-pass transmembrane protein highly expressed in vascular endothelial cells (ECs), where it plays fundamental roles in EC functions. ENG is implicated in several cardiovascular disorders including hereditary haemorrhagic telangiectasia, pulmonary arterial hypertension (PAH) and preeclampsia. However, molecular mechanisms underlying ENG function are not fully understood. Initially identified as a co-receptor for TGF-β signalling, ENG's extracellular domain was later found to only bind BMP9 and BMP10 with high affinity. The relationship between these two observations is unclear. Here, we provide evidence for two primary functions of co-receptor ENG. First, ENG efficiently displaces prodomains from BMP9 and BMP10, enabling effective capturing of both ligands from the circulation. Second, ENG binds to and recruits TGFBRII into the BMP9 signalling complex, thereby explaining ENG's involvement in both TGF-β and BMP9 pathways. We identify BMP9 target genes *NOG* and *ADAMTSL2* as preferentially dependent on ENG and show that their transcript levels have strong positive correlation with *ENG* in human lung tissues; the expression levels of all three genes are significantly reduced in PAH. Our findings address an important gap in our understanding on ENG biology and provide crucial insight for therapeutic targeting these pathways in vascular diseases.

Endoglin (ENG) is a single-pass transmembrane protein highly expressed in vascular endothelial cells (ECs). It has a large extracellular domain (ECD) and a small intracellular domain (ICD). Loss-of-function mutations in *ENG* cause type I hereditary haemorrhagic telangiectasia (HHT1)[1]; symptoms include telangiectasia in the nose, gastrointestinal tract and skin, as well as larger arteriovenous malformations in the brain, lungs and liver, which can cause life-threatening haemorrhage. ENG mutations are also found in patients with pulmonary arterial

[1]The Victor Phillip Dahdaleh Heart & Lung Research Institute, School of Clinical Medicine, University of Cambridge, Cambridge Biomedical Campus, Cambridge, UK. [2]Institute of Metabolic Science, MRC Metabolic Diseases Unit, University of Cambridge, Addenbrooke's Hospital, Cambridge, UK. [3]European Molecular Biology Laboratory, European Bioinformatics Institute (EMBL-EBI), Wellcome Genome Campus, Cambridge, UK. [4]Department of Cell and Chemical Biology, Leiden University Medical Centre, Leiden, The Netherlands. [5]Division of Pulmonary and Critical Care Medicine, University of California, Los Angeles, CA, USA. [6]National Heart and Lung Institute, Imperial College London, London, UK. [7]Centro de Investigacion Principe Felipe, Valencia, Spain. [8]Biosciences Institute, International Centre for Life, Newcastle University, Newcastle Upon Tyne, UK. ✉e-mail: wl225@cam.ac.uk

hypertension (PAH)[2], a vascular disorder characterised by remodelling and occlusion of small pulmonary vessels, leading to increased right ventricular systolic pressure and ultimately right-sided heart failure. ENG ECD can be shed from the cell surface and released into circulation as soluble ENG (sENG). Concentrations of sENG are markedly elevated by more than tenfold in the maternal circulation of women with preeclampsia (PE)[3], and thought to contribute to its pathogenesis[4]. PE is a pregnancy-specific hypertensive disorder which increases the risk of poor outcomes for both mother and baby. Circulating sENG is also elevated in PAH (by about twofold) and has been proposed as a biomarker for the prognosis of PAH[5]. Despite the important roles of ENG and sENG in different cardiovascular diseases, the molecular mechanisms of ENG function and its role in the pathogenesis of HHT, PAH and PE remain elusive.

ENG was discovered initially as a component of the transforming growth factor-β (TGF-β) family signalling complex[6]. TGF-β family ligands, including bone morphogenetic proteins (BMPs), are primarily homodimers, initiating cellular responses by forming a signalling complex with two copies of a type I receptor and two copies of a type II receptor. TGF-β type I receptor (TGFBRI), also termed Activin receptor-like kinase 5 (ALK5), and TGF-β type II receptor (TGFBRII) mediate signals from TGF-β1, TGF-β2 and TGF-β3. Using cell surface labelling with radioactive ligands followed by crosslinking and immunoprecipitation in human umbilical vein ECs (HUVECs), ENG was found to associate with TGF-β1 and TGF-β3 but not TGF-β2[6]. Hence, sENG was proposed as a ligand trap for TGF-β1 and TGF-β3[4]. Moreover, it has been shown that TGF-β1 can signal through both ALK1 and ALK5 in ECs to regulate endothelial activation state and that its signalling through ALK1 requires ENG[7–9]. ENG was reported to interact with TGFBRII and ALK5 via both its ECD and ICD in COS-7 cells transiently transfected with plasmids containing the TGF-β receptors and ENG[10,11]. Of note, all these findings were published before 2007.

In 2007, ALK1 was found to be a specific type I receptor for BMP9 and BMP10[12,13]. In 2011, direct binding assays using recombinant sENG-Fc showed that ENG ECD binds with high affinities only to BMP9 and BMP10, not to any other TGF-β family receptors or ligands[14]. Thus, ENG is a co-receptor for BMP9 and BMP10 signalling, and sENG was thought to be a ligand trap for BMP9 and BMP10 but not for TGF-βs[14]. However, these findings cannot explain the reports prior to 2007, and a gap in knowledge remains and needs to be addressed to understand the function of ENG fully.

BMP9 and BMP10 are synthesised as pre-proproteins. Upon secretion, the prodomain is cleaved but remains non-covalently bound to the growth factor domain (GFD) forming pro:BMP9 and pro:BMP10 (Supplementary Fig. 1a, b). Crystal structures revealed that the prodomains bind to BMP9 and BMP10 overlapping with the type II receptor binding site[15–18] (Supplementary Fig. 1b–d). Although pro:BMP9 and pro:BMP10 signal with equal potency as their respective GFDs in ECs[17], signalling requires displacement of the prodomain. Unprocessed proBMP9 and proBMP10, where the prodomain cannot be displaced are not active[19,20]. The crystal structure of the ENG orphan domain (ENG-OR) bound to BMP9 further shows that its binding site on BMP9 also overlaps with both the prodomain and the type II receptor binding sites[21] (Supplementary Fig. 1e). How ENG functions as a co-receptor to facilitate BMP9 and BMP10 signalling has remained unclear.

To fully appreciate the critical role of ENG in ECs and in vascular biology, it is important to understand its function at the molecular level. We first revisited sENG. We have previously shown that sENG in the circulation is mainly monomeric and does not inhibit BMP9 and BMP10 signalling in vascular ECs[22]. However, at the elevated concentrations found in PE plasma, sENG can inhibit BMP9 and BMP10 signalling in Eng knockout ECs or in non-ECs with low ENG[22,23]. Building on this, the present study investigates how ENG resolves the overlapping binding site with the prodomain and the type II receptors

to facilitate BMP9 signalling. We show that ENG efficiently displaces the prodomain from both pro:BMP9 and pro:BMP10 at sub-molar ratios in solution, far more effectively than the type II receptors. Furthermore, ENG binds to and recruits TGFBRII into the BMP9-ALK1 complex, enabling TGFBRII to participate in BMP9 signalling. Comparative transcriptomics in Eng knockout and control cells identified Nog (encodes Noggin) and Adamtsl2 as BMP9 target genes that are more dependent on ENG, a finding confirmed in human dermal microvascular ECs (HDMECs), which express low levels of ENG. Importantly, mRNA expression of ENG, NOG and ADAMTSL2 is strongly correlated in human lung tissues and significantly reduced in PAH patients' lungs compared to healthy subjects. Together, our results support a model in which ENG enables BMP9 signalling through alternative type II receptors, with TGFBRII participating in BMP9 signalling in an ENG-dependent manner. This model unifies earlier important findings of ENG's role in TGF-β receptor complexes with later reports of ENG's direct interaction with BMP9 and BMP10, establishing ENG as a key mediator of BMP9 and TGF-β crosstalk in ECs. This study reconciles the long-standing controversy of ALK1- and ALK5-mediated TGF-β signalling in controlling the activation states of vascular ECs. ENG mutations were identified in HHT and PAH over two decades ago; yet no curative therapies are available for these vascular disorders. Research progress on ENG function has been slow due to this knowledge gap. Our findings provide a mechanistic framework that can accelerate ENG research and guide therapeutic targeting ENG in vascular disease.

## Results

### ENG ECD, but not the type II receptor ECDs, can effectively displace BMP9 and BMP10 prodomains

The prodomains of BMP9 and BMP10 need to be displaced before a signalling complex can be assembled (Supplementary Fig. 1b–d). We have previously shown that binding of sENG to pro:BMP9 displaces its prodomain[22]. Here, we investigated this further by asking whether ENG could displace the prodomain from pro:BMP9 at a sub-molar ratio and whether this also applies to pro:BMP10.

We purified full-length ENG ECD in both dimeric (sENG(D)) and monomeric (sENG(M)) forms[22] and performed native PAGE analysis, which separates proteins and protein complexes according to size and charge. Supplementary Fig. 2a summarises the molecular weight and the isoelectric point (pI) of all proteins and complexes used in this study.

Pro:BMP9 separated on native PAGE to yield a strong pro:BMP9 complex band (band 1) and a faint prodomain band (band 2) (Fig. 1a). When pre-mixed with increasing amounts of sENG, additional bands were observed representing sENG(M):BMP9 (bands 3, 4 and 5) and sENG(D):BMP9 (bands 6, 7 and 8) complexes. The identities of bands 1 to 8 were confirmed by excision and re-running on an SDS-PAGE (Supplementary Fig. 2b) and the relative abundance of the complexes was quantified (Supplementary Fig. 2c–e). Time-course experiment showed no evidence of prodomain reassociation after displacement (Supplementary Fig. 2f, g). Of note, sENG can displace the prodomain and form a complex with BMP9 GFD at molar ratios as low as 0.3, the lowest sENG concentration tested.

We then performed analytical gel filtration to confirm the native PAGE results (Fig. 1b, c). Under the identical running condition, pro:BMP9 eluted in fraction 18. When pre-mixing with 0.3-fold of sENG(M) (Fig. 1b) or 0.3-fold of sENG(D) (Fig. 1c), we detected additional peaks of sENG(M):BMP9 complex (fraction 16 in Fig. 1b) and sENG(D):BMP9 complex (fraction 13 in Fig. 1c), accompanied by the free prodomain (fraction 20 in Fig. 1b, c).

Similar results were obtained for pro:BMP10. On native PAGE (Fig. 1d), 0.3-fold sENG, either monomeric or dimeric, efficiently displaced the prodomain from pro:BMP10, resulting in reduced intensity of the pro:BMP10 band (band 2), the increased intensity of the BMP10

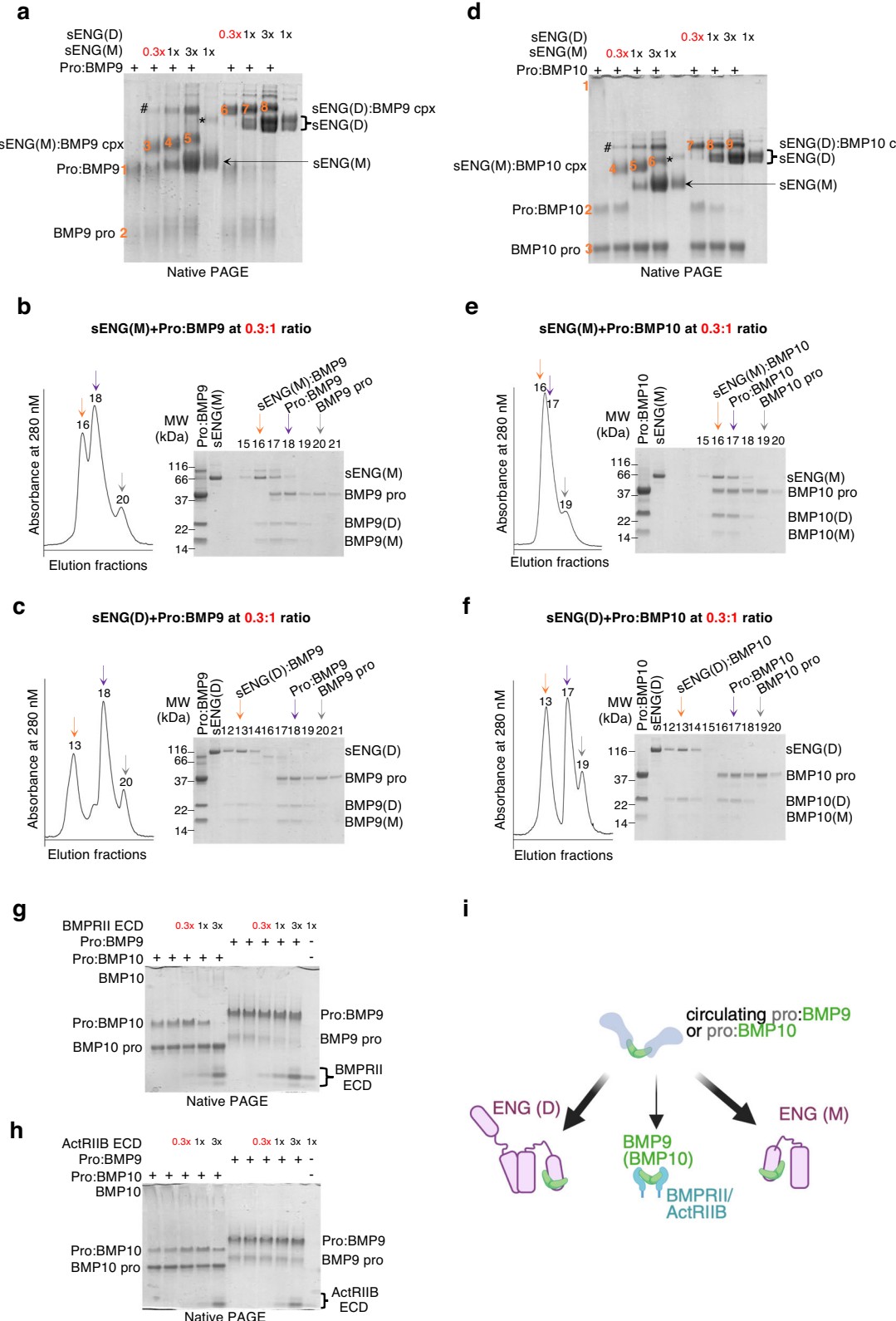

prodomain band (band 3), and the appearance of new complex bands (bands 4-6 for sENG(M):BMP10 and bands 7–9 for sENG(D):BMP10). The identities of bands 1 to 9 were confirmed (Supplementary Fig. 3a), and complex abundance was quantified (Supplementary Fig. 3b–d). Time-course experiments showed no reassociation after prodomain displacement (Supplementary Fig. 3e, f). Similar to BMP9, 0.3-fold

sENG(M) or sENG(D) displaced the prodomain and formed the corresponding complex with BMP10 GFD on gel filtration (Fig. 1e, f).

We next asked whether the type II receptor ECDs could displace the prodomain. It has been reported that BMP receptor type II (BMPRII)-Fc and activin receptor type IIB (ActRIIB)-Fc bind to BMP9 and BMP10 with the highest affinity[15]. Thus, we examined the ability of

**Fig. 1 | sENG(D) and sENG(M) can displace the prodomain from pro:BMP9 and pro:BMP10 complexes at sub-molar ratio. a–c** sENG can displace the prodomain from the pro:BMP9 complex at a ratio as low as 0.3:1, as shown by native-PAGE (**a**), and gel filtration (**b, c**). The identities of protein bands highlighted in orange numbers in (**a**) have been confirmed by excision and re-running on SDS-PAGE (Supplementary Fig. 2b). **d–f** sENG can displace the prodomain from the pro:BMP10 complex at as low as 0.3:1 ratio, showing by native-PAGE (**d**), and gel filtration (**e, f**). The identities of protein bands highlighted in orange numbers in (**d**) have been confirmed by excision and re-running on SDS-PAGE (Supplementary Fig. 3a). Bands

marked with # are likely the BMP9 or BMP10 in complex with sENG(D) due to the small amount of sENG(D) in the sENG(M) prep (marked with *). **g, h** Native-PAGE of prodomain displacement assay by BMPRII (**g**) and ActRIIB (**h**) ECDs. **i** Summary diagram (generated using BioRender) illustrating that prodomain from pro:BMP9 and pro:BMP10 can be displaced effectively by ENG (thick arrow) but not so effectively by the type II receptors (thin arrow). Experiments were repeated for 3 times (**a, d, g, h**) and representative gels are shown. sENG(D) dimeric sENG, sENG(M) monomeric sENG, Pro:BMP9 (or 10) prodomain bound BMP9 (or 10), pro prodomain, cpx complex, ECD extracellular domain.

BMPRII ECD and ActRIIB ECD to displace the prodomain. Using untagged monomeric BMPRII ECD and ActRIIB ECD, we did not detect stable complexes of BMPRII or ActRIIB with BMP9 or BMP10 GFD on size exclusion chromatography under physiological buffer conditions such as PBS. We then performed native PAGE analysis under the same conditions as above. In the presence of BMPRII ECD (Fig. 1g), we did not detect any prodomain displacement for pro:BMP9, because pro:BMP9 bands remained at the same intensity across all experimental conditions (Supplementary Fig. 3g). In contrast, the pro:BMP10 band started to decrease when BMPRII was at a 1:1 ratio to pro:BMP10, suggesting partial displacement of the prodomain. Complete displacement was observed when BMPRII ECD was at threefold in excess. ActRIIB could not displace the prodomain of either pro:BMP9 or pro:BMP10 under identical experimental condition (Fig. 1h and Supplementary Fig. 3h).

These results confirm that both sENG(M) and sENG(D) displace the prodomain from pro:BMP9 and pro:BMP10 complexes far more efficiently than the type II receptors (Fig. 1i), suggesting that a significant function of ENG is to capture the circulating pro:BMP9 and pro:BMP10 at the cell surface.

## The stoichiometry of ENG interactions with BMP9 and BMP10

To understand how ENG facilitates BMP9 and BMP10 signalling, it is important to elucidate the binding mode and stoichiometry of their interactions. Crystal structures were reported for ENG-OR alone, ENG-OR in complex with BMP9, and ENG zona pellucida (ZP) domain alone[21]. ENG-OR in complex with BMP9 shows that two copies of the orphan domain bind at the two knuckle areas of the BMP9 dimer, overlapping with the type II and prodomain binding sites (Supplementary Fig. 1e). Of note, to solve the ENG-OR and BMP9 structure, the complex was generated by co-expressing the maltose binding protein (MBP)-orphan domain fusion protein with BMP9 GFD in the same cells. Therefore, this crystal structure did not address how ENG displaced the prodomain.

As both sENG(D) and sENG(M) can completely displace the prodomain effectively and form complexes with BMP9 (and BMP10), there are a total of six possible forms of complexes after mixing pro:BMP9 (or pro:BMP10) with sENG(D) or sENG(M) (Fig. 2a). Based on the crystal structures[21], it was modelled that one ENG dimer binds to one BMP9 dimer with BMP9 dimer being clamped by the two protomers of sENG(D) (Fig. 2a, form I). ENG monomer in complex with BMP9 was not detected in this report, probably due to the differentially tagged ENG employed and/or the limitation of the co-expression system.

We first assessed the size of the complexes by analytical gel filtration chromatography. sENG(M) was separately premixed with pro:BMP9 or pro:BMP10 at 6× molar excess, or sENG(D) premixed with pro:BMP9 or pro:BMP10 at 3× molar excess for 30 min before being loaded onto an S200 10/300 gel filtration column. As shown in Fig. 2b, c, control runs showed that sENG(M), sENG(D), pro:BMP9 and pro:BMP10 eluted at peak positions 1, 2, 3 and 7, respectively. Despite being at 6× molar excess, sENG(M) formed a complex with BMP9 or BMP10 GFD dimer at a 1:1 ratio (peaks 4 and 8) because no species that are close to or larger than sENG(D) was detected on the gel filtration, eliminating the possibility of two sENG(M) binding to the same GFD

(Fig. 2a, form VI). Similarly, at 3× molar excess, the complexes between sENG(D) with BMP9 (peak 6) or BMP10 (peak 10) were only slightly shifted to the left of the sENG(D) peak, suggesting it is unlikely to have two copies of sENG(D) binding to the same GFD (Fig. 2a, form IV). Thus, gel filtration narrows down the possible forms of the complexes from six to four (Fig. 2a, forms I, II, III and V). Further experiments are required to ascertain whether forms I, II and III are all possible and whether one of them is a preferred form.

We also used crosslinking followed by SDS-PAGE to evaluate the stoichiometry. If two sENG(M) can bind to the same GFD dimer, we would expect a crosslinked species with a molecular weight larger than sENG(D) (Fig. 2d). Similarly, if two sENG(D) can bind to the same GFD dimer, the resulting species would be greater than 200 kDa (Fig. 2e). Experimental data revealed that crosslinked sENG(M) with BMP9 or BMP10 were detected at a molecular weight slightly bigger than sENG(M) but well below sENG(D) (Fig. 2f, g, black arrows), and crosslinked sENG(D) with BMP9 or BMP10 were detected at molecular weight below 200 kDa (Fig. 2f, g, arrowheads). Thus, the crosslinking experiments confirmed the gel filtration results that one molecule of sENG(M) or sENG(D) binds preferentially to only one BMP9 or BMP10 GFD dimer.

## Binding of ENG to BMP9 and BMP10 reduces their binding affinity for ALK1

Since the ENG binding site on BMP9 partially overlaps with the type II but not the type I receptor binding site, we questioned whether ENG binding to BMP9 or BMP10 would affect their binding affinity for the type I receptor ALK1. To test this, surface plasmon resonance (SPR) experiments were performed by flowing BMP9 and BMP10 GFDs, with or without preincubation with sENG, over a CM5 chip immobilised with ALK1-Fc (Fig. 3). BMP9 bound strongly to ALK1, with a $K_D$ value of 58.6 pM (Fig. 3a, d), consistent with previous reports[15,17]. Preincubation with sENG(M) reduced BMP9 affinity to ALK1 by threefold, primarily due to a slower association rate (Fig. 3b, d). Incubation with sENG(D) reduced its affinity to ALK1 by 7.2-fold, due to a combination of a slower association rate and a slightly faster dissociation rate (Fig. 3c, d). A similar scenario was observed for BMP10. BMP10 bound to ALK1-Fc with a $K_D$ value of 183 pM (Fig. 3a, d). Binding to sENG(M) led to a ninefold decrease in affinity due to a slower association rate (Fig. 3b, d). Further reduction of the affinities (16-fold) was observed upon binding to sENG(D), due to both a slower association rate and a faster dissociation rate (Fig. 3c, d).

In conclusion, in the presence of ENG, BMP9 and BMP10 bind to ALK1 with slightly reduced affinity. This is likely due to steric hindrance because slower association rates primarily drive the reduced affinity. Of note, the presence of ALK1 ECD does not affect the binding affinities of BMP9 or BMP10 to ENG ECD (Supplementary Fig. 4).

Since BMP9 and BMP10 showed differences in prodomain displacement by BMPRII ECD (Fig. 1g and Supplementary Fig. 3g), and sENG:BMP10 complexes have different binding profiles to ALK1-Fc compared to sENG:BMP9 complexes (Fig. 3b, c), we reasoned that ENG and BMPRII may have different roles in BMP9 and BMP10 signalling. In this report, we focused the remainder of the mechanistic study on BMP9 only.

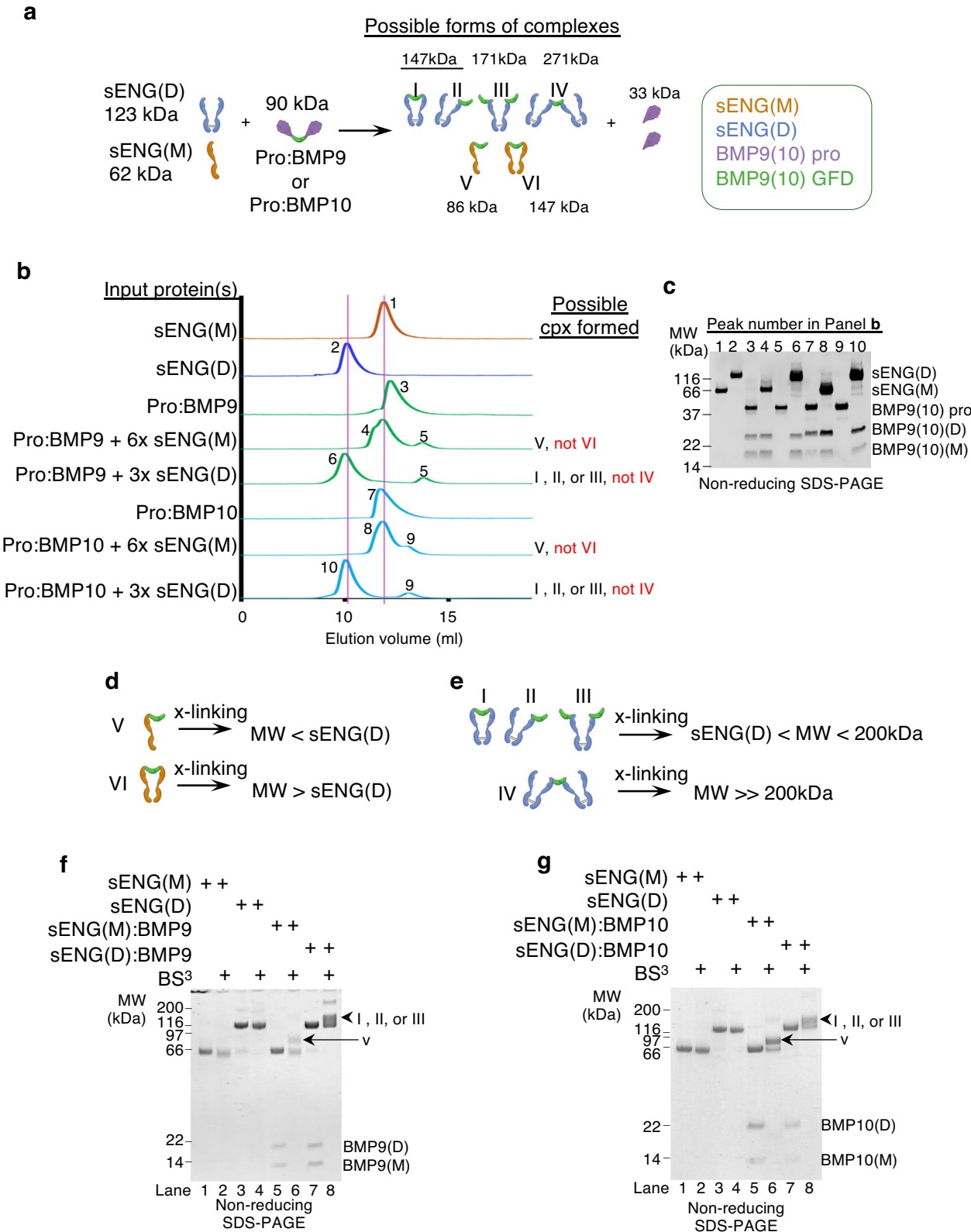

**Fig. 2 | The binding stoichiometry between sENG and BMP9 (or BMP10).**
**a** Schematic diagram depicting the prodomain displacement from pro:BMP9 or pro:BMP10 by sENG(D) and sENG(M) and the formation of the six possible forms of the complexes. **b, c** Analysis of different complexes by analytical gel filtration. Gel filtration chromatographs (**b**) show the elution positions of different proteins and complexes. The two pink lines indicate the elution positions for sENG(D) and sENG(M), respectively. Fractions at the positions labelled 1 to 10 were run on non-reducing SDS-PAGE (**c**) to confirm their identities. **d–g** Analysis of binding stoichiometry by crosslinking. **d, e** Schematic diagrams of the crosslinking experiment for sENG(M):BMP9 (or BMP10) (**d**) and sENG(D):BMP9 (or BMP10) (**e**). **f, g** Crosslinking and non-reducing SDS-PAGE of sENG:BMP9 complexes (**f**) and sENG:BMP10 complexes (**g**). Experiments were repeated for two (**g**) or three times (**b, c, f**). pro prodomain, cpx complex, GFD growth factor domain, x-linking crosslinking.

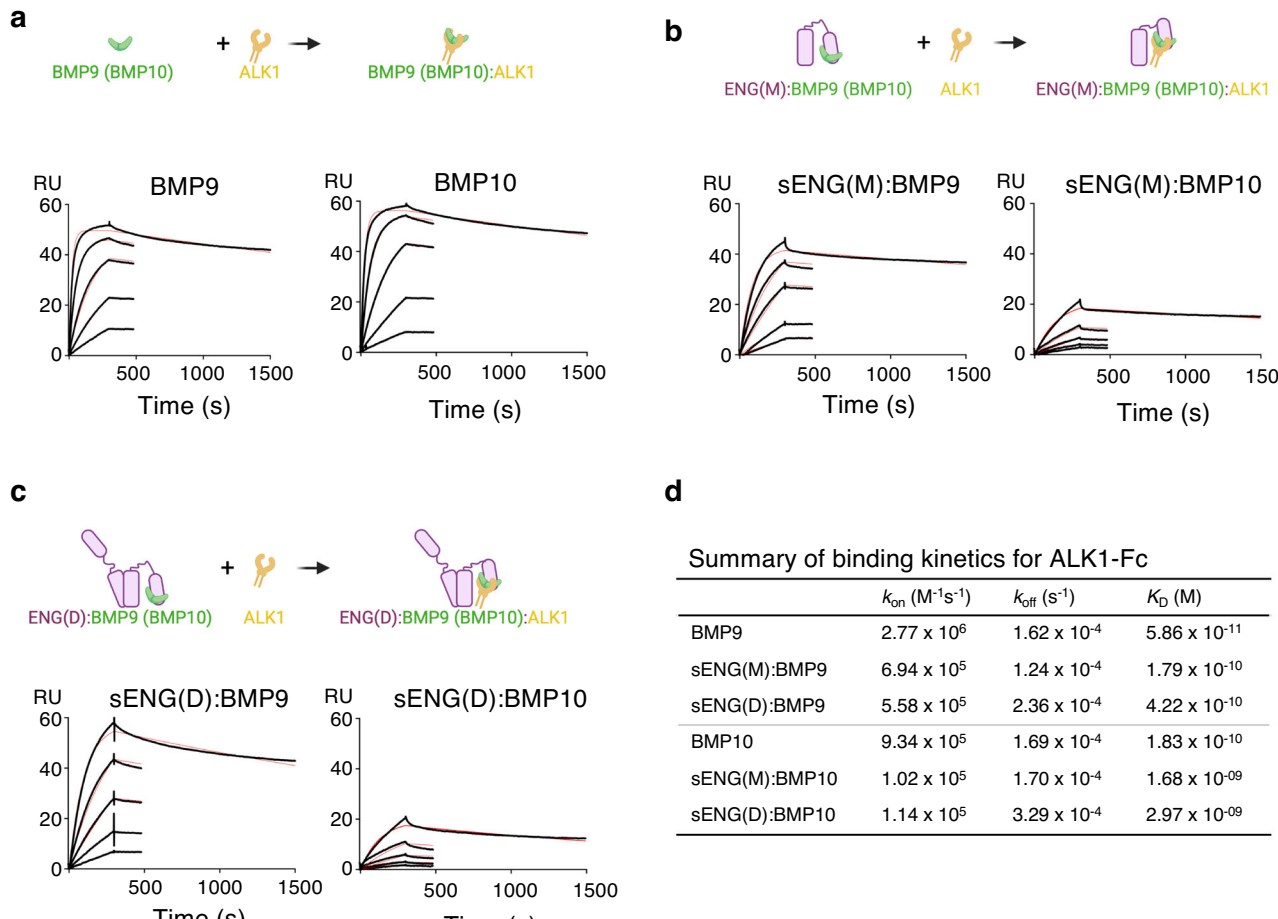

**Fig. 3 | Binding of BMP9 and BMP10 to ALK1-Fc in the absence and presence of sENG(M) and sENG(D).** **a–c** Diagrams and SPR sensorgrams of BMP9 and BMP10 binding to ALK1-Fc (**a**), sENG(M):BMP9 and sENG(M):BMP10 binding to ALK1-Fc (**b**) or sENG(D):BMP9 and sENG(D):BMP10 binding to ALK1-Fc (**c**). BMP9 or BMP10 were pre-incubated with sENG(M) or sENG(D) at 1:1.2 ratio for 30 min at room temperature to ensure complex formation before applying to the flow cells immobilised with ALK1-Fc. BMP9 was applied from 17.77 nM to 0.22 nM in threefold serial dilutions. BMP10 was applied from 53.33 nM to 0.66 nM in threefold serial dilutions. **d** Summary of the binding kinetics. All measurements are the averages from two runs.

## BMP type II receptors cannot efficiently displace ENG from ENG:BMP9 complex

We next investigated whether, once ENG has captured BMP9, the type II receptors are able to displace ENG to form a signalling complex. Using monomeric receptor ECDs, we did not detect displacement of ENG from its complex with BMP9 by BMPRII or ActRIIB ECDs in either native PAGE or gel filtration assays. We developed an ELISA-based competition assay to measure the displacement of ENG from sENG:BMP9 complex (Fig. 4a). Interestingly, BMPRII and ActRIIB showed displacement of sENG from sENG(M):BMP9 complex only at 125-fold excess, whereas BMPRII ECD displaced a small proportion of sENG from sENG(D):BMP9 complex from fivefold molar excess, reaching over 50% displacement when BMPRII ECD was at 125-fold molar excess. ActRIIB ECD displaced sENG from the sENG(D):BMP9 complex from 25-fold excess (Fig. 4b).

## Type II receptors for ENG-dependent BMP9 signalling

We have previously shown that sENG:BMP9 complex can signal in wild-type ECs effectively, but not in *Eng* knockout ECs[22]. Since BMPRII and ActRIIB ECDs cannot effectively displace sENG from its complex with BMP9, the obvious other candidate to capture BMP9 from their complex with sENG is the cell surface ENG. This also suggests that after circulating pro:BMP9 is captured by cell surface ENG, it cannot effectively handover the ligand to cell surface BMPRII or ActRIIB for signalling. Another mechanism might be in place to allow ENG-bound BMP9 to signal.

Crystal structures revealed that all TGF-β family type II receptors interact with ligands at the knuckle epitope apart from TGFBRII, which binds TGF-β ligands at the fingertip[15,18,24–26]. Interestingly, ENG was initially discovered as a TGF-β co-receptor[6] and was detected in a complex with TGFBRII in [125]I-TGF-β ligand crosslinking assays[27]. Overlaying the ENG-OR:BMP9 structure[21] onto the TGFBRII:TGF-β1 structure[28] revealed that TGFBRII could potentially contact ENG:BMP9 complex without any clash (Fig. 5a). Therefore, we asked the question whether TGFBRII could participate in BMP9 signalling via ENG.

We performed siRNA knockdown of *TGFBR2* in human pulmonary artery ECs (HPAECs) in parallel with knockdown of other type II receptors known to mediate BMP signalling and investigated pSmad1/5 signalling responses to BMP9 treatments. In agreement with previous findings[29], knocking down the type II receptor individually did not completely abolish BMP9 signalling (Fig. 5b, c). However, knocking down *TGFBR2* or *ACVR2A* resulted in partial reduction of the signalling, whereas knocking down *BMPR2* did not affect BMP9-induced Smad1/5 phosphorylation, suggesting TGFBRII could be one of the type II receptors mediating BMP9 signalling.

We next asked whether TGFBRII can interact with BMP9 extracellularly to mediate BMP9 signalling. We did not detect any direct

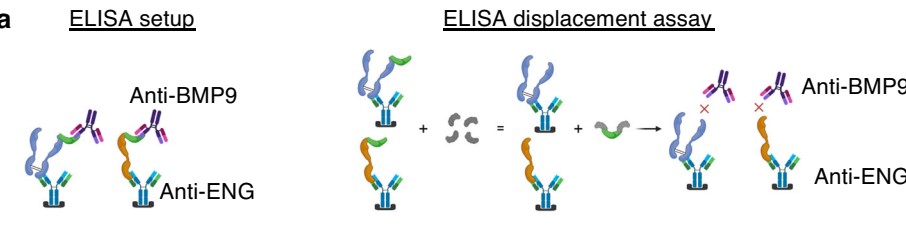

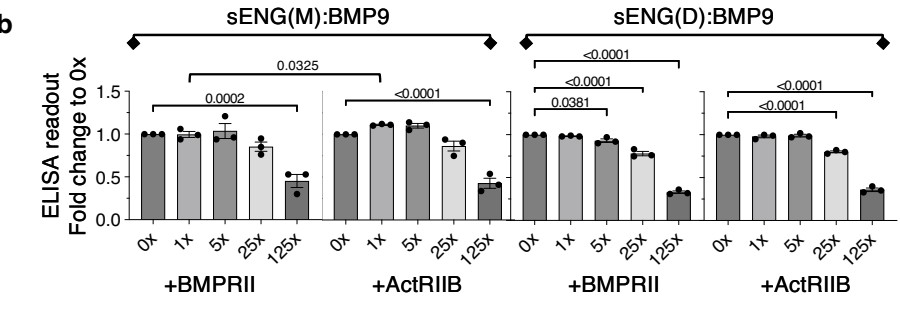

**Fig. 4 | Type II receptors cannot effectively displace ENG from the sENG:BMP9 complexes. a** Diagram illustrating the ELISA-based displacement assay. Using anti-ENG antibody as the coating antibody and anti-BMP9 as the detecting antibody, only the sENG:BMP9 complexes can be measured. If the type II receptors displace sENG from the sENG:BMP9 complexes, a reduction of ELISA signal would be detected. For illustration purpose, only one out of the three possible sENG(D):BMP9 forms is included here. **b** Fixed concentrations of sENG(M):BMP9 or sENG(D):BMP9 was incubated with increasing concentrations of monomeric BMPRII ECD or ActRIIB ECD at molar ratios as indicated before applying to the ELISA measurement. $N = 3$ independent experiments. Data were normalised to the control samples without type II receptor and shown as means ± SEM. One-way ANOVA, with Dunnett's post-tests to compare with 0× receptor ECD in each experiment. To evaluate the displacement efficiencies between BMPRII and ActRIIB, unpaired, two-sided $t$-tests were applied to compare between BMPRII and ActRIIB at the same molar excess. Only significant $p$-values ($<0.05$) are shown.

interaction between BMP9 GFD and TGFBRII ECD using either SPR or ELISA-based binding assays (Supplementary Fig. 5a–d). However, since it has been shown that ENG interacts with TGFBRII intracellularly by immunoprecipitation in transfected COS-7 cells[10], we asked whether such intracellular interactions are present in human primary ECs, which could enable ENG to bring TGFBRII to ALK1 via BMP9. We have previously generated an anti-ENG affinity column, which pulled down sENG from ex vivo-cultured placenta-conditioned media[22]. Using this affinity column, we pulled down endogenous ENG and its interaction partners from total cell lysates of human primary aortic ECs (HAOECs), HUVECs as well as HPAECs and analysed total eluates by mass spectrometry. A BSA-coupled NHS column was used as the negative control to allow detection of those proteins that interact with the column non-specifically (Fig. 5d). ECs under three different culturing conditions were harvested and analysed: (1) medium with 0.1% Fetal Bovine Serum (FBS); (2) medium containing 10% FBS; (3) medium with 0.1% FBS and pro:BMP9. Of note, cells in 0.1% FBS medium do not have active BMP9 signalling, whereas 10% FBS contains high concentrations of BMP9 as well as other serum factors. Figure 5e summarises the mass spectrometry results, showing as the total number of unique peptide-counts of ENG binders as well as controls. The detected peptides are highlighted on the protein sequences in Fig. 5f and Supplementary Fig. 6. The anti-ENG column pulled down many more ENG fragments from all ECs across three medium conditions (23–32 fragments) compared with the control BSA column (3–9 fragments). Interestingly, under 0.1% FBS condition, both TGFBRII and ALK1 were detected only in the anti-ENG column elutes for all the ECs examined. This indicates that endogenous ENG, TGFBRII and ALK1 are associated in human primary ECs under this condition. Under 10% FBS condition and upon BMP9 treatment, where Smad/1/5 phosphorylation is activated

(Fig. 5g), we detected more peptides from TGFBRII, but no ALK1 peptide was detected. As a positive control, we detected a previously reported ENG binding protein TRIM21 in all anti-ENG column eluates[30]. For a negative control, RALA was pulled down from both anti-ENG and BSA columns at a similar level. Of note, under 10% FBS and BMP9-treated conditions, ENG may still interact with ALK1 indirectly via BMP9 (Fig. 3), but we will not be able to detect such interaction in this pull-down assay because the anti-ENG antibody used for the affinity column is TRC105, which blocks BMP9 binding to ENG[21,31]. Proteomic findings are summarised in schematics (Fig. 5h). These data support the idea that ENG can bring TGFBRII and ALK1 together, allowing TGFBRII to participate in BMP9 signalling in an ENG-dependent manner.

## Model of ENG-dependent and ENG-independent BMP9 signalling

Our data strongly suggest that one aspect of ENG's co-receptor function is to capture BMP9 and BMP10 from the circulation and deliver them to cell surface ALK1. ENG ECD monomer is over 60 kDa (561 amino acids) with two distinct domains, whereas ALK1 ECDs is a much smaller (96 amino acids) single domain protein (Supplementary Fig. 7a). ENG clearly has the advantage of having a better reach into the circulation lumen for capturing BMP9 and BMP10, but at the same time, it will need to have a flexible structure to allow it to deliver the bound ligands to ALK1. We noticed that sENG(D) migrated as two distinct bands on the native PAGE (Fig. 1a, d), and both could bind to ligands because both bands shifted upon binding to BMP9 or BMP10 (Fig. 1a, d). This suggests that native sENG(D) may exist in two different forms with different net charge or shape. Although gel filtration could not separate the two forms and sENG(D) runs under a single peak (Fig. 2b), crosslinking experiments showed that only about 50% of the

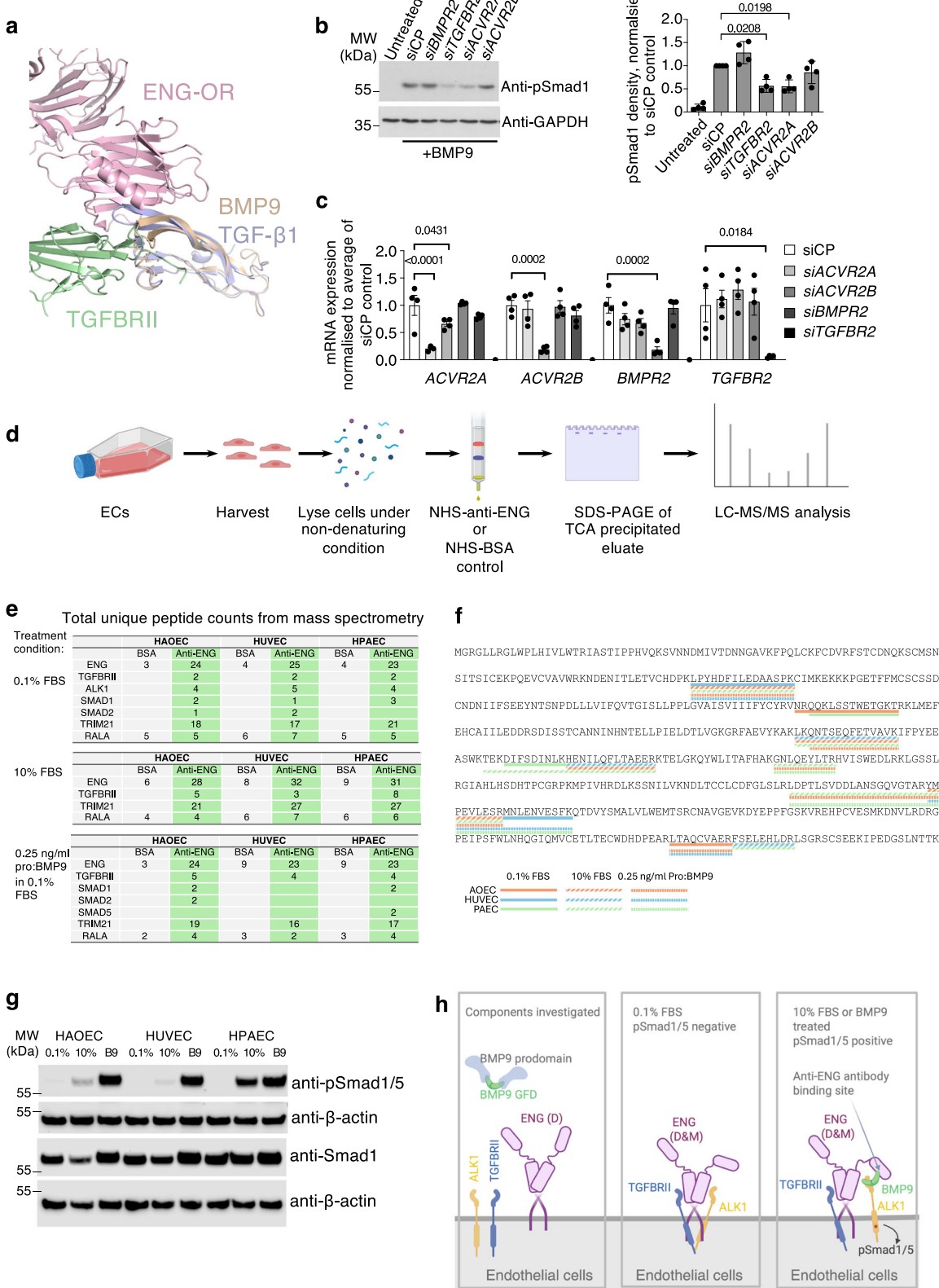

sENG(D) were crosslinked in solution, despite that four crosslinkers with different spacer arm length were used across different crosslinker concentrations (Fig. 6a, b and Supplementary Fig. 7b).

To further develop the model for cell surface ENG function, we need to know whether it exists in only dimeric form or both dimer and monomer forms. We noticed that in total cell lysates of human primary ECs from different vascular beds and different donors, a monomeric form of ENG can be consistently detected regardless of the culturing media (Supplementary Fig. 7c, d). To examine whether such monomeric ENG is at the cell surface or intracellular, we performed surface biotinylation coupled with immunoprecipitation by streptavidin beads (Fig. 6c) to detect ENG only at the cell surface. After surface biotinylation, both dimeric and monomeric ENG were pulled down by streptavidin agarose beads in both HPAECs and HUVECs (Fig. 6d, lane 7). For

**Fig. 5 | ENG recruits TGFBRII to BMP9 signalling complexes. a** Superposition of the ENG-OR:BMP9 (5HZW) structure with the TGF-β1:TGFBRII (3KFD) structure by the ligands, showing ENG and TGFBRII can contact BMP9 simultaneously without any clash. Figure prepared using Pymol (The PyMOL Molecular Graphics System, Version 2.0, Schrödinger). **b** Effects of type II receptor knockdown on Smad1 phosphorylation induced by BMP9 in human pulmonary arterial endothelial cells (HPAECs). Cells were serum-starved in EGM-2, 0.1% FBS overnight, and treated with BMP9 for 15 min before being harvested for immunoblotting against an anti-pSmad1 antibody. GAPDH was used as the loading control. $N = 4$ independent siRNA experiments using the HPAECs from two donors. The quantification of pSmad1 relative to siCP control is shown on the right. **c** RT-qPCR examining the efficiency of siRNA in (**b**) and expression of other type II receptors after siRNA treatment. For (**b** and **c**), means ± SEM are shown, one-way ANOVA with Dunnett's post-tests to compare with siCP-treated samples, for each receptor (**c**). Only significant $p$-values (<0.05) are shown. **d** Schematic description of the ENG pull-down assay. NHS = N-Hydroxysuccinimide, here refers to the HiTrap NHS-activated high-performance column. **e** Unique peptide counts from the mass spectrometry of the ENG pull-down assay. Only selected target proteins and controls are shown. The sequences of the peptides are mapped onto the protein sequences, as shown in (**f**) for TGFBRII and Supplementary Fig. 6. **g** Immunoblots showing Smad1/5 phosphorylation status of the samples used in the mass spectrometry experiments. **h** Diagrams (generated using BioRender) summarising the components investigated and the protein-protein interactions identified in the pull-down/mass spectrometry experiments. HAOECs human aortic endothelial cells, HUVECs human umbilical vein endothelial cells.

controls, no ENG was pulled down without biotinylation (Fig. 6d, lane 4); and the intracellular protein β-actin was not labelled by cell surface biotinylation, therefore only existed in the unbound fractions (Fig. 6d, lane 5), not in the elution (Fig. 6d, lane 7). These results support that both dimeric and monomeric ENG are present at the surface of these human primary ECs (Fig. 6e).

Our results thus support the following model of ENG co-receptor function for BMP9 signalling in ECs (Fig. 6f). Cell surface ENG can be in both monomeric and dimeric forms. Both can efficiently capture the BMP9 GFDs from circulating pro:BMP9, displace the prodomains, deliver the GFDs to cell surface ALK1 and form a ternary complex with BMP9 and ALK1. A proportion of this complex can dissociate to allow BMPRII or ActRIIA to bind; however, in the remaining complex, ENG-bound TGFBRII can phosphorylate ALK1 to initiate Smad-dependent or Smad-independent signalling.

## The difference between ENG-dependent and ENG-independent signalling

We next questioned whether ENG-dependent and ENG-independent signalling paths would result in similar or different signalling outcomes; for example, different type II receptors may have different efficiency in phosphorylating ALK1. Previous efforts using microarray to compare sENG(M):BMP9 and pro:BMP9 signalling in HPAECs did not reveal any difference[22], probably due to the significant overlap in signalling. Here, we isolated mouse lung ECs (MLECs) from *Eng*-floxed mice (*Rosa26-Cre^ERT2;Eng^fl/fl*) and depleted *Eng* by 4-OH-tamoxifen treatment (Fig. 7a). Because only ENG-independent signalling occurs in *Eng^-/-* (KO) cells, by comparing sENG(M):BMP9 signalling in *Eng^+/+* (Ctrl) cells versus pro:BMP9 signalling in KO cells, we anticipated better signals to tease out ENG-dependent signalling (Fig. 7b).

We confirmed the EC identity of Ctrl and KO MLECs by CD31 and VE-Cadherin staining (Fig. 7c and Supplementary Fig. 8a). The knockout of *Eng* was confirmed by ENG staining, immunoblotting and qPCR (Fig. 7c, d and Supplementary Fig. 8a, b). As we have shown previously, knocking out *Eng* led to reduced BMP9 signalling in the Smad1/5 phosphorylation assay (Fig. 7d)[22]. We treated Ctrl cells with sENG(M):BMP9 (or PBS) and KO cells with pro:BMP9 (or PBS) and then performed RNA sequencing to explore *Eng*-dependent and *Eng*-independent target genes. We first validated our RNA sequencing results by confirming the known differentially expressed genes (DEGs) induced by BMP9 in Ctrl cells and KO cells. Treatment of sENG(M):BMP9 in Ctrl cells (Fig. 7e) and pro:BMP9 in KO cells (Fig. 7f) induced canonical BMP9 target genes, such as *Smad6*, *Smad7*, *Id1* and *Atoh8*. We then compared the DEGs in PBS-treated Ctrl and KO cells to examine the effect of knocking out *Eng* on MLECs at baseline. Only three genes were significantly downregulated after losing ENG, which are *Col3a1*, *Elovl6* and *Reln*. However, the fold changes were minimal (Log2FC = 0.45 to 0.69) (Fig. 7g). In contrast, when comparing DEGs from ligand-treated Ctrl cells and KO cells, we detected 76 significantly changed DEGs (Fig. 7h). Several DEGs had fold changes equal or bigger than 2, with the most significant fold changes observed for *Nog* and *Adamtsl2*, suggesting that the induction of these two genes was more dependent on ENG.

Significantly changed DEGs in Fig. 7h were subject to pathway analysis using String[32]. Three distinct clusters of protein-protein interaction network were detected (Supplementary Fig. 9a), centred around Irf9/Ifih1/Oasl1, Col3a1/Col6a2/Col6a1, and Bmp4, respectively. Gene ontology and KEGG pathway showed that significantly enriched terms include response to interferon-β, regulation of the viral processes, and TGF-β signalling pathways (Fig. 7i, j and Supplementary Fig. 9b, c).

## Validation of ENG-dependent target genes in mouse and human primary ECs

RT-qPCR experiments were performed to validate the significantly changed DEGs (Fig. 7h). We first measured the basal expression of selected genes relative to the housekeeping gene *Hprt* and calculated the ratio of the basal expression in KO/Ctrl cells (Fig. 8a, dark grey bars). Most of the values are close to 1, indicating that the basal expressions of these genes were similar in KO and Ctrl cells without BMP9 treatment. We then measured the fold induction of these genes after ligand treatment (ENG(M):BMP9 in Ctrl cells and pro:BMP9 in KO cells, as shown in Fig. 7b), and calculated their ratio of KO/Ctrl (Fig. 8a, light grey bars). Interestingly, the ratios for most of the genes are below 1, indicating that the fold induction in KO cells is lower than in the Ctrl cells, consistent with the RNAseq data (Fig. 7h). When comparing the ratios in untreated cells (Fig. 8a, dark grey bars) with those in ligand-treated cells (Fig. 8a, light grey bars) for each investigated gene, the differences in *Adamtsl2* and *Nog* were statistically significant. We confirmed that the lower fold induction in *Adamtsl2* and *Nog* was not due to the difference in basal expression (Fig. 8b) but because of the lower fold induction in the KO cells (Fig. 8c).

It has been reported that HDMECs express significantly lower ENG than ECs from other vascular beds[33]. We hypothesised that HDMECs might be similar to the *Eng* KO MLECs in that they would have less fold induction of ENG-dependent BMP9 target genes. We first compared ENG expression in HDMECs with that in HAOECs and HUVECs at both protein and mRNA levels (Fig. 8d–f) and confirmed that ENG expression in HDMECs was much lower than that in HAOECs and HUVECs. We then performed BMP9 signalling assays in these cells, at both 0.1 and 1 ng/ml BMP9 GFD concentrations. BMP9-induced the canonical BMP target gene *ID1* at both concentrations in all three primary ECs, and fold induction was very similar across the three ECs (Fig. 8g). In contrast, BMP9 effectively induced *NOG* and *ADAMTSL2* in both HAOECs and HUVECs, but not in HDMECs (Fig. 8g), and this was not due to the difference in the basal expression of these genes in different ECs (Fig. 8h). This is consistent with *NOG* and *ADAMTSL2* being ENG-dependent genes, and the lack of induction in HDMECs is probably due to the very low ENG expression.

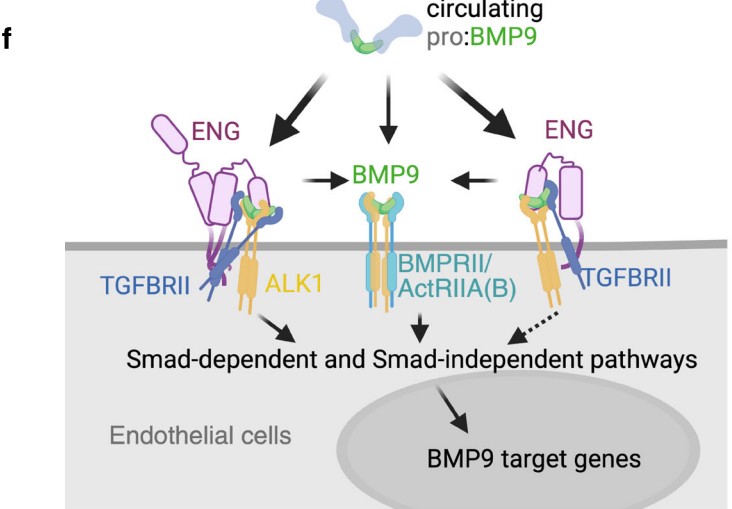

**a**

-DTT  +DTT
MW - + - + DSG
(kDa)
116—
66—
Crosslinking distance: 7.7 Å

-DTT  +DTT
MW - + - + BS(PEG)₅
(kDa)
116—
66—
Crosslinking distance: 21.7 Å

-DTT  +DTT
MW - + - + BS³
(kDa)
116—
66—
Crosslinking distance: 11.4 Å

-DTT  +DTT
MW - + - + BS(PEG)₉
(kDa)
116—
66—
Crosslinking distance: 35.8 Å

**b**

A proportion of ENG(D) cannot be crosslinked (x-linked)

No crosslinker | With crosslinker

ENG (D) ENG (D) | X-linked / not X-linked: ENG (D) ENG (D)

Endothelial cells

**c**

Primary ECs → Surface biotinylation → Total cell lysate → Streptavidin beads → PBS wash → Elution → SDS-PAGE and anti-ENG immunoblot / Anti-β-actin as a control

**d**

-Biotin  +Biotin
Total lysate / Unbound / Wash / Elution / Unbound / Wash / Elution

MW (kDa)
HPAEC
130—
95—    ENG(D) / ENG(M)    anti-ENG
55—    anti-β-actin

MW (kDa)
HUVEC
130—
95—    ENG(D) / ENG(M)    anti-ENG
55—    anti-β-actin

Lane 1 2 3 4 5 6 7
Non-reducing SDS-PAGE

**e**

Both ENG(D) and ENG(M) can be detected at cell surface

ENG (D) ENG (M)

Endothelial cells

**f**

circulating pro:BMP9

ENG    BMP9    ENG

TGFBRII / ALK1    BMPRII/ActRIIA(B)    TGFBRII

Smad-dependent and Smad-independent pathways

Endothelial cells

BMP9 target genes

### ENG, NOG and ADAMTSL2 in PAH patient samples

To assess the human relevance of our findings, we analysed a publicly available microarray dataset of human lung tissue[34] and found the expression levels for *ENG*, *NOG* and *ADAMTSL2* were all significantly reduced in PAH patients' lungs compared to control (Fig. 9a). Importantly, the expression levels of both *NOG* and *ADAMTSL2* were strongly and positively correlated with *ENG*, and with each other (Fig. 9b). This is consistent with our finding that *NOG* and *ADAMTSL2* are *ENG*-dependent. We next examined these genes in the published whole blood RNAseq dataset from 363 PAH samples versus 126 controls[35]. The expression levels for *ENG* and *NOG* were both significantly reduced in PAH and significantly correlated with each other (Fig. 9c, d). Furthermore, lower levels of *NOG* mRNA were associated with poorer transplant-free survival in PAH patients (Fig. 9e) and worse REVEAL 2.0

**Fig. 6 | Working model of ENG-mediated BMP9 signalling on vascular ECs. a, b** A proportion of sENG(D) cannot be crosslinked. **a** Crosslinking of sENG(D) (1.15 μM) by different crosslinkers (final concentrations of 1 mM) with crosslinking distances from 7.7 to 35.8 Å. Samples were run on SDS-PAGE after crosslinking. Two independent experiments were performed and representative gels are shown. **b** Diagram interpreting the crosslinking result in (**a**) that a proportion of ENG(D) is flexible or in a different local conformation, therefore cannot be crosslinked. **c** Flowchart of the cell surface biotinylation assay to examine cell surface ENG. **d** Non-reducing SDS-PAGE and immunoblots against ENG or β-actin of different samples from the cell surface biotinylation assay in both HPAECs and HUVECs. **e** Diagram summarising the results from (**d**). **f** Model depicting the co-receptor function of ENG in BMP9 signalling. The majority of circulating prodomain-bound BMP9 can be effectively captured by cell surface ENG. While some ENG-bound BMP9 may dissociate to allow BMPRII or ActRIIA to bind, ENG can interact with TGFBRII and with ALK1 via BMP9, allowing TGFBRII to participate in BMP9 signalling in an ENG-dependent manner. Although this model applies for both dimeric and monomeric ENG, monomeric ENG has only one copy of ENG to interact with TGFBRII; and the type II receptor ECDs do not displace sENG from sENG(M):BMP9 as effectively as from sENG(D):BMP9 complexes as shown in Fig. 4b, hence ENG(M) is predicted to be less efficient to facilitate BMP9 signalling than ENG(D). For this reason, a dashed line is drawn for ENG(M)-mediated signalling. For illustration clarity, only one of the three possible sENG(D):BMP9 forms in Fig. 2a is depicted here.

Lite mortality risk scores[36] (Spearman test, Rho = −0.14, $P$ = 0.0083). In a published proteomics dataset of whole blood plasma[37,38], Noggin protein levels were also significantly reduced in PAH (Fig. 9f), consistent with our findings. Of note, we observed increased sENG protein in PAH patients' plasma as previously reported[5].

## Discussion
In this study, we report how ENG participates in BMP9 and BMP10 signalling as a co-receptor in ECs despite binding to the ligands at overlapping site with the prodomain and the BMP type II receptors. We show that ENG can displace the BMP9 and BMP10 prodomains at sub-molar ratio, enabling efficient capture of circulating BMP9 and BMP10. Secondly, ENG does not need to dissociate from BMPs but can help assemble a signalling complex by binding to TGFBRII directly and to ALK1 via BMP9 or BMP10. Thirdly, we identified *Nog* and *Adamtsl2* as two BMP9 target genes that are preferentially dependent on ENG, and the mRNA expression of *NOG* and *ADAMTSL2* in human lungs is strongly correlated with that of *ENG*. Our results provide a comprehensive model of how BMP9 signals in ECs. While ALK1 is absolutely required for all BMP9 and BMP10 signalling, there is a significant degree of redundancy in the type II receptor utilisation, with TGFBRII also playing a role in BMP9 signalling in addition to the previously reported BMPRII and ActRIIA.

Our data has an important impact on reconciling two previous controversial findings on ENG and on BMP9/TGF-β signalling in ECs. ENG was discovered as a component of the TGF-β signalling complex[6] using radiolabelled TGF-β1 treated cells followed by crosslinking. In contrast, direct protein binding assays[14] later showed that sENG did not bind to any of the TGF-β family ligands or receptors apart from BMP9 and BMP10. As a result, a persistent disconnection between these findings has raised questions on how ENG contributes to both TGF-β and BMP9 signalling, creating a controversy that has also slowed progress in understanding the molecular mechanism of ENG function. Our data have filled this long-standing knowledge gap by demonstrating that ENG interacts with TGFBRII in un-transfected human primary ECs, and that this interaction enables TGFBRII to participate in ALK1-mediated BMP9 signalling. The reason ENG was shown as positive in the original $^{125}$I-TGF-β1 label-crosslinking experiments[6] can now be explained by its interaction with TGFBRII intracellularly, and the crosslinker Disuccinimidyl suberate (DSS) being membrane permeable, therefore permitting intracellular crosslinking. This also explains the observation that ENG showed a similar TGF-β binding profile to TGFBRII, i.e., binding with high affinities to TGF-β1 and TGF-β3 but not TGF-β2[6], and ENG binding to TGF-β is dependent on TGFBRII[39,40].

Both ECD and ICD of ENG have been reported to interact with TGFBRII in transiently transfected COS-7 cells[10]. Although we have not detected direct interaction between BMP9 and TGFBRII ECD in our in vitro direct binding assays (Supplementary Fig. 5), it remains a possibility that ENG and TGFBRII ECDs could make contact in the proper cellular context when other components are present, such as ligands or other cell surface proteins. Structural analysis shows that ENG, TGFBRII and ALK1 ECDs all bind to ligands at different sites (Supplementary Fig. 10), and there will not be any structural constraint extracellularly for ENG to bring TGFBRII and ALK1 together.

In ECs, ALK1 is required for BMP9 signalling irrespective of which type II receptors are present; hence knocking down ALK1 can completely abolish BMP9-induced Smad1/5 phosphorylation. In contrast, the type II receptors for BMP9 signalling have a significant degree of promiscuity and redundancy, and we generally only see 20–50% of the reduction after knocking down the type II receptors individually when cells are treated with low concentrations of BMP9 for a short period, such as 10–15 min. In a previous study, si*ACVR2A* and si*BMPR2* individually in HPAECs did not cause significant loss of BMP9 signalling (BMP9 treatment at 1 ng/ml for 1 h), whereas knocking down both receptors only led to approximately 50% decrease of Smad1/5 phosphorylation[29]. Interestingly, BMP9 signalling was reduced by around 50% in *Eng*$^{-/-}$ MLECs[22], suggesting that both ENG-dependent and ENG-independent pathways play significant roles in BMP9 signalling.

Our model supports the above observation in siRNA experiments, suggesting that the overall signalling outcome is crucially dependent on the local receptor expression levels and other cellular context, such as whether any high affinity ligands are present to compete with the type II receptor binding[41]. Our model also suggests that for TGFBRII to interact with ENG and play a meaningful role in BMP9 signalling, it will need to be expressed at a similar level to ENG. Indeed, single cell RNAseq data from the Human Protein Atlas showed that in vascular ECs, *TGFBR2* is highly expressed and at a similar level to *ENG*, slightly higher than *BMPR2* and *ACVRL1*, and at an order of magnitude higher level than *TGFBR1* (Supplementary Fig. 11).

It is intriguing that we only pulled down ALK1 from ECs in 0.1% FBS condition when signalling is off, and not from ECs in 10% FBS or 0.1% FBS + BMP9 where BMP9 signalling is active and ALK1 is activated. Interestingly, it has been reported that ALK5 can only interact with ENG in an inactive form[10]. We also pulled down SMAD1, SMAD2 and SMAD5 under different conditions and in different cell types. This could be due to their association with the type I receptors. Mixed R-SMADs complex has been described before[42] and we cannot exclude the possibility of the interaction of SMAD1 with SMAD2 in the 0.1% FBS condition. We consistently detected the interaction between endogenous ENG and TGFBRII in all three medium conditions, regardless of whether BMP9 signalling was active or not. It is interesting to note that in the previous study in transfected COS-7 cells, it was also found that TGFBRII remains associated with ENG in both its active and inactive forms[10].

Human genetics strongly suggests that dysregulated endothelial BMP signalling involving BMP9/ALK1/BMPRII/ENG plays a significant role in endothelial dysfunction and is likely to be the initial trigger for the pathogenesis of PAH[43,44]. Interestingly, mutations in *TGFBR2* were found in ECs of plexiform lesions from PAH patients. Microsatellite site mutations and reduced protein expression of TGFBRII were found in 6 out of 19 lesions[45]. In immunochemistry staining, 31 of 35 (89%) PAH plexiform lesions did not express TGFBRII, whereas ECs within PAH pulmonary arteries not containing plexiform lesions and pulmonary

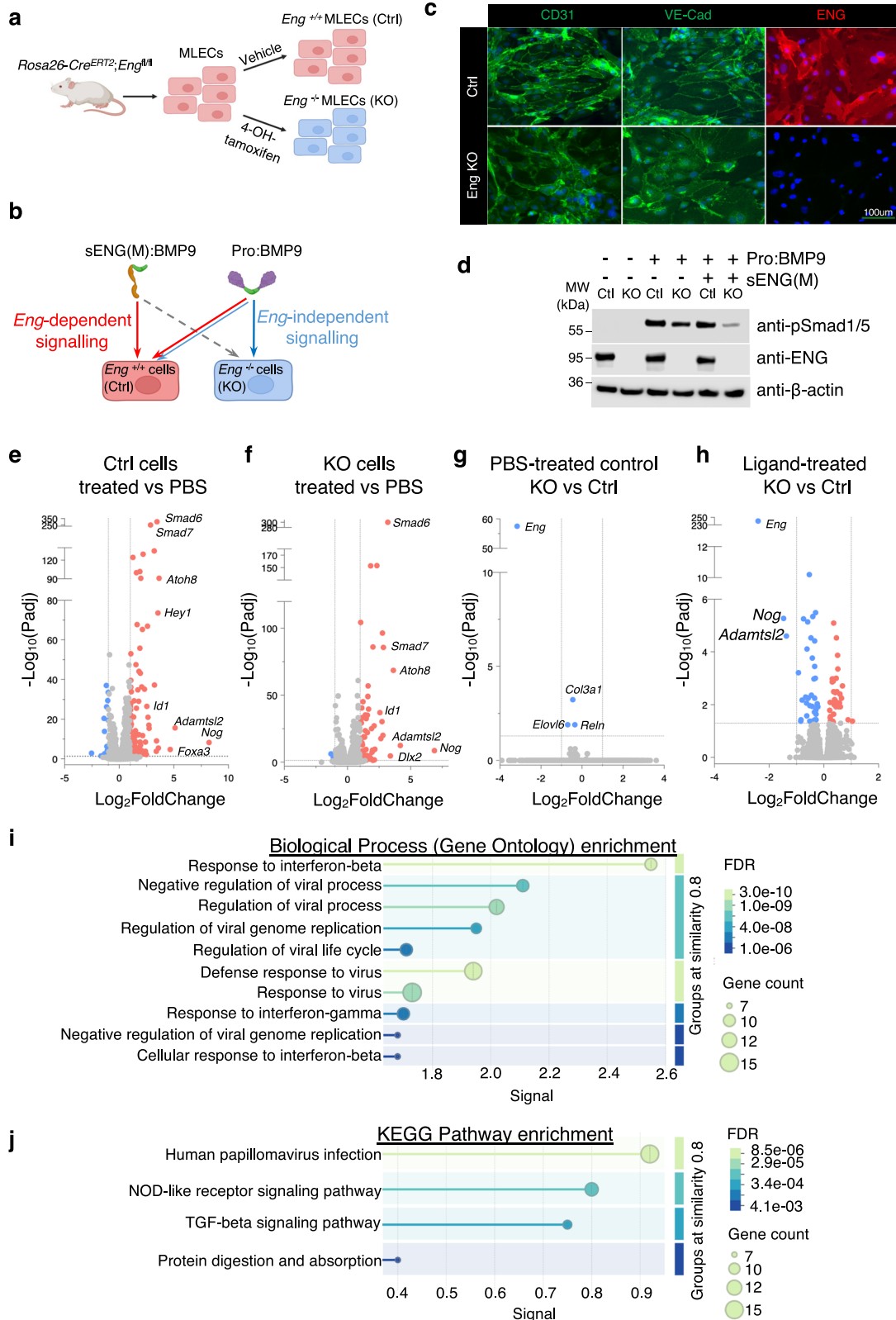

arteries in 4 normal lungs expressed TGFBRII abundantly. This supports that loss of endothelial TGFBRII contributes to endothelial dysfunction in PAH, consistent with our finding that TGFBRII plays a role in BMP9 signalling.

We identified *Nog* and *Adamtsl2* as two BMP9 target genes that are preferentially dependent on ENG. Interestingly, both are extracellular regulators of TGF-β family signalling. ADAMTSL2 has been proposed

to be involved in regulating TGF-β bioavailability and its mutations were found in an autosomal recessive disorder Geleophysic dysplasia[46]. Noggin is an inhibitor for many BMP ligands, including BMP4[47] and BMP7[48], but not BMP9 or BMP10[49].

Expression of ENG can also be detected in non-ECs where ALK1 is expressed at low levels or not expressed and BMP9 can signal through ALK2. Whether the mechanisms of ENG function we identified here

**Fig. 7 | ENG-dependent and ENG-independent BMP9 signalling. a** Diagram illustrating the generation of *Eng*[+/+] (control or Ctrl) and *Eng*[-/-] (Knockout or KO) mouse lung ECs (MLECs). **b** Strategy for comparing ENG-dependent and ENG-independent signalling. We have previously shown[22] that sENG:BMP9 complex has compromised signalling in KO cells due to the loss of cell surface ENG (grey dashed arrow). Hence, its signalling in Ctrl cells is mostly ENG-dependent (red arrow). In contrast, pro:BMP9 can signal in Ctrl cells through both ENG-dependent and ENG-independent pathways, but in KO cells only through ENG-independent pathway (blue arrow). Therefore, the comparison of sENG(M):BMP9 signalling in Ctrl cells with pro:BMP9 signalling in KO cells will reveal the maximum signal to detect the difference in ENG-dependent and ENG-independent BMP9 signalling. **c** CD31 and VE-Cadherin staining of Ctrl and KO MLECs to confirm their EC identity, as well as ENG staining to show the knockout of *Eng*. Quantification of staining from three independent mouse lines shown in Supplementary Fig. 8a. **d** BMP9 signalling in Ctrl

and KO MLECs. *Eng* KO cells have reduced Smad1/5-dependent signalling in response to pro:BMP9 treatment compared with control cells, and in the excess of sENG(M), KO cells almost lose Smad1/5 mediated signalling. The middle panel confirms *Eng* knockout by anti-ENG immunoblot. Results from one experiment from five independent repeats are shown. **e–h** Volcano plots showing the differentially expressed genes (DEGs) from the RNA sequencing results. **e** Ctrl MLECs, sENG(M):BMP9-treated versus PBS control; **f** KO MLECs, pro:BMP9-treated versus PBS control; **g** PBS-treated, KO versus Ctrl MLECs; **h** Ligand-treated, KO versus Ctrl MLECs. The horizontal dotted line has Y = 1.3, indicating where Padj = 0.05. Two dashed vertical lines have X = ±1, indicating where fold changes = 2. **i, j** Pathway analysis of the significantly changed DEGs in (**h**) using the String Sever. The biological processes of Gene Ontology (**i**) and KEGG pathway enrichment (**j**) are shown. String network connections, WikiPathways enrichment and Monarch enrichment are shown in Supplementary Fig. 9.

apply to the role of ENG in other cell types and with other type I receptors are yet to be investigated. It is intriguing to note the strong positive correlations of *NOG* and *ADAMTSL2* gene expression with *ENG* in human lungs, which contain both ECs and non-ECs. Moreover, although ECs do not contribute to whole blood mRNA, we also observed a weak yet significant positive correlation between *ENG* and *NOG*, suggesting at least some aspects of the ENG mechanism identified here could apply to non-ECs and other type I receptors.

We and others[50] have demonstrated that serum Noggin protein levels are reduced in PAH patients. We now demonstrate that in whole blood RNAseq, both *ENG* and *NOG* expression levels are reduced, and a lower Noggin level correlates with worse survival in PAH patients. These observations on *ENG* and *NOG* in blood samples might be used as PAH biomarkers that are linked to the TGF-β/BMP pathway, the key dysregulated pathway in PAH identified by human genetics.

In summary, we show that a primary role of ENG as a co-receptor is to capture circulating BMP9 and BMP10 effectively from plasma. ENG can bring TGFBRII into the BMP9 signalling complex. We identified BMP9 target genes *Nog* and *Adamtsl2* as preferentially dependent on ENG and further validated this finding in human primary ECs and in lung tissues. We provide a model of the ENG function and BMP9 signalling, which reconciles different historical reports on the ENG function. Our data fill an important gap in our knowledge of ENG function and provide a framework for strategies targeting BMP9/ALK1/ENG/BMPRII pathways for therapeutic applications.

## Materials
All chromatography columns were purchased from GE Healthcare/Cytiva. TRC105 antibody was a kind gift from TRACON. HMEC-1 and HEK EBNA cells were from American Type Culture Collection (ATCC). HPAECs, HUVECs, HAOECs and HDMECs were purchased from Lonza and PromoCell. BMP9 and BMP10 GFDs, sENG-Fc, ALK1-Fc were purchased from R&D Systems.

## Methods
### Protein expression and purification
cDNAs of full-length human BMP9 and BMP10, including the prodomain, sENG (Residue 1-586)-(His)$_6$ were cloned into a *pCEP4* plasmid. For protein expression, each plasmid was transfected into HEK EBNA cells in Dulbecco's modified Eagle Medium (DMEM, 11965092, Gibco), 5% FBS. For BMP10 expression, human full-length FURIN cDNA in *pCEP4* was co-transfected to facilitate processing. The transfection medium was replaced by CD CHO (10743-029, Thermo Fisher Scientific) medium on day 2, and the conditioned medium was harvested twice a week for 5 harvests.

To purify pro:BMP9 and pro:BMP10, the conditioned medium was loaded onto a 5-ml HiTrap Q HP column (17115401, Cytiva) and eluted with a NaCl gradient. This was followed by a HiLoad Superdex 200 pg 16/600 gel filtration column pre-equilibrated in 20 mM Tris, 50 mM NaCl, pH 7.4. Pooled fractions were loaded on a MonoP 5/50 GL

column and eluted with a NaCl gradient. The eluted sample was polished using a HiLoad Superdex 200 pg 16/600 gel filtration column pre-equilibrated in 20 mM Tris, 150 mM NaCl, pH 7.4.

sENG-His was expressed as a mixture of sENG(D) and sENG(M)[22]. They were initially captured on a 5-ml HisTrap Excel column (17371206, Cytiva) and the eluted fractions were pooled and dialysed in 20 mM Tris, 50 mM NaCl, pH 7.4. The dialysed sample was then loaded on a 5-ml HiTrap Q HP column and eluted with a NaCl gradient. sENG(D) and sENG(M) were separated by a HiLoad Superdex 200 pg 16/600 gel filtration column in 20 mM Tris, 150 mM NaCl, pH 7.4.

The cDNA for human ALK1 (Residue 22–118), BMPRII (Residue 27–150) and ActRIIB (Residue 24–115) ECDs were cloned separately into a *pET-39b* plasmid (70090, Novagen) to create a fusion protein DsbA-(His)$_6$-TEV site-ECD. The expression and purification for all the ECDs were identical. In brief, the plasmid for each protein was transformed into *E. coli* Rosetta DE3 (70954, Novagen) for protein expression. At mid-log phase, cells were induced by 200 μM isopropyl β-D-thiogalactopyranoside (IPTG) and then grown at 22 °C overnight. The ECDs were extracted from periplasmic fractions according to the pET System Manual (Novagen). The His-tagged fusion proteins were captured and eluted from a 5-ml HisTrap excel column. This was followed by overnight TEV digestion and dialysis in TBS at 4 °C. The digested proteins were passed through the HisTrap excel column to remove the His-tagged DsbA and TEV protease. The ECDs were then passed through a HiLoad Superdex 75 pg 16/600 gel filtration column to purify the proteins further.

### Cell surface biotinylation
HPAECs and HUVECs were seeded in 6-well plates overnight in endothelial growth medium 2 (EGM2, C-22111, PromoCell) with 10% FBS. Cells were washed 3 times with ice-cold PBS pH 8.0 and then incubated with 0.33 mg/ml Thermo EZ-Link Sulfo-NHS-SS-Biotin (21328, Thermo Fisher Scientific, membrane impermeable) in PBS for 30 min at 4 °C. The reaction was quenched with 2 ml per well 35 mM Tris pH 8.0 for 10 min at room temperature, followed by 2 times wash with PBS. The cells were lysed with 200 μl per well lysis buffer (125 mM Tris pH 7.4, 10% glycerol, 2% SDS, 2 mM EDTA and 1× protease inhibitor (11836170001, Sigma-Aldrich)) on ice. The lysate was centrifuged at 20,000 × g for 10 min at 4 °C and the supernatant was incubated with 150 μl Pierce Avidin Agarose (20219, Thermo Fisher Scientific) and incubated with mixing for 1 h at room temperature. The resin was then washed 5 times with PBS and boiled in 1× SDS loading dye. ENG was assessed by immunoblotting. Anti-ENG (555690, BD Bioscience) and anti-β-Actin (A5441, Sigma-Aldrich) were used and related information can be found in Supplementary Data 1.

### Crosslinking
Proteins were incubated with 1 mM (final concentration) of DSG, BS[3], BS(PEG)$_5$ or BS(PEG)$_9$ (A35392, A39266, A35396, 21582, respectively, from Thermo Fisher Scientific) in PBS pH 7.4–8.0 for 45 min at room

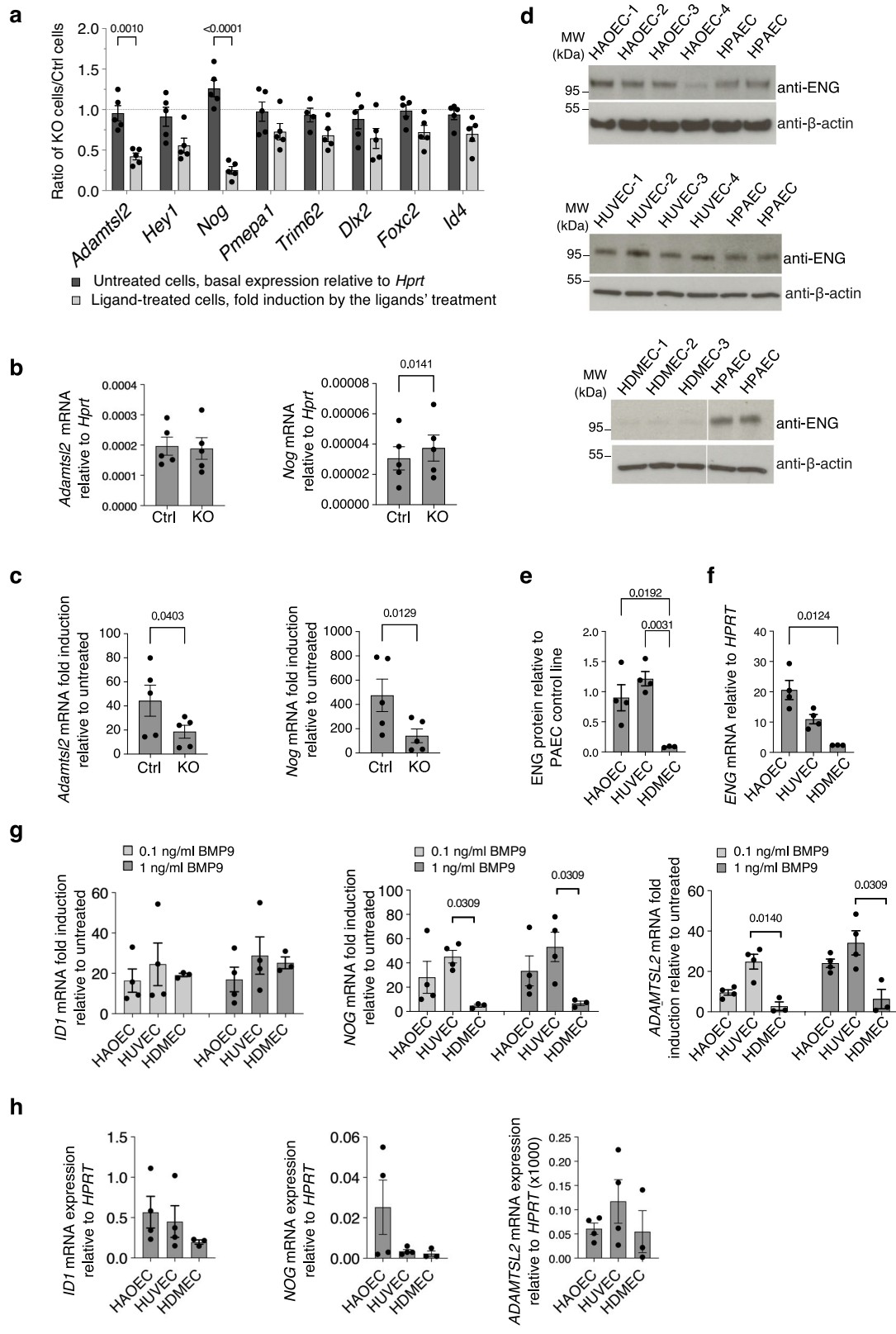

temperature. The reaction was then quenched with 40 mM Tris (final concentration) pH 7.4–8.0 for 15 min at room temperature. Concentrations of the crosslinkers were titrated to optimise the cross-linking efficiency.

**Native and SDS-PAGE**

Pro:BMP9 or pro:BMP10 was pre-mixed with sENG(D), sENG(M), BMPRII or ActRIIB ECDs at ratios indicated in the figure legends in TBS in a final volume of 8 µl and incubated for 30 min at room temperature.

**Fig. 8 | Validation of DEGs from RNAseq in MLECs and human primary ECs.**
**a**–**c** Validation of ENG-dependent BMP9 target genes in MLECs. $N = 5$ biological
repeats. Cells were treated as in Fig. 7b (0.2 ng/ml BMP9 GFD). **a** Basal expression of
selected DEGs was measured by RT-pPCR and the ratios in *Eng* KO cells to Ctrl cells
(dark grey bars) were plotted. Fold induction by the BMP9 ligands' treatments was
measured and ratios of KO/Ctrl cells were plotted (light grey bars). The dotted line
with Y = 1 indicates that there is no difference between KO and Ctrl cells. **b** Basal
expression of *Adamtsl2* and *Nog* in KO and Ctrl MLECs, respectively. **c** Fold
induction of *Adamtsl2* and *Nog* after the treatment with the BMP9 ligands in KO and
Ctrl MLECs, respectively. **d**–**h** Validation in human primary ECs. HAOECs human
aortic endothelial cells, HUVECs human umbilical vein endothelial cells, HDMECs
human dermal microvascular endothelial cells. $N = 4$ for HAOECs and HUVECs, $N = 3$
for HDMECs, all biological repeats. **d**–**f** ENG expression in different ECs at protein

(**d**, **e**) and mRNA (**f**) levels. Cells were serum-starved in EGM-2 with 0.1% FBS
overnight before being harvested for mRNA expression and immunoblotting. In
(**d**), the same human pulmonary artery ECs (HPAEC) lysate was loaded on each gel
twice to allow quantification across different immunoblots. In the HDMEC blot, two
parts of the same gel are shown. **g** Fold induction of *ID1*, *NOG* and *ADAMTSL2* mRNA
levels induced by pro:BMP9 in different human primary ECs, at 0.1 and 1 ng/ml GFD
concentrations, respectively. All treatments were 1.5 h. **h** Basal expression of *ID1*,
*NOG* and *ADAMTSL2* mRNA levels in different human primary ECs. Means ± SEM are
shown (**a**–**c**, **e**–**h**). Statistical tests are as follows: two-way ANOVA mixed-effect
model comparing the ratio of basal expression with the ratio of fold induction for
each gene (**a**), two-sided paired *t*-tests (**b**, **c**), Kruskal–Wallis tests to (**e**, **f**, **h**), and to
each ligand concentration group in (**g**), followed by Dunn's multiple comparisons.
Only significant *p*-values (<0.05) shown.

The samples were then run on a 10% native PAGE at 4 °C. Where
appropriate, the Coomassie InstantBlue (ab119211, Abcam) stained
native gel bands were cut out and inserted directly into the wells of a
12% SDS-PAGE gel and run under non-reducing condition to reveal
their identity.

## Surface plasmon resonance
Surface plasmon resonance (SPR) binding experiments were per-
formed using a Biacore T200 system (Cytiva/GE Healthcare). For all
assays, ALK1-Fc or sENG-Fc was immobilised onto a Series S CM5 chip
(BR100530, Cytiva) via amine coupling. The SPR analytes were injected
in duplicate at a flow rate of 40 μl/min (in a buffer containing 10 mM
HEPES pH 7.4, 150 mM NaCl, 3.4 mM EDTA, 0.5 mg/ml BSA and 0.005%
(v/v) surfactant P20) at 25 °C. The surface was regenerated with 1–3
injections (depending on the binding partners) of 4 M Guanidine
Hydrochloride. The kinetic rate constants were obtained by fitting the
corrected data to a 1:1 interaction model or steady-state model using
BIAevaluation software (Cytiva/GE Healthcare).

## Generating *Eng*-iKO ECs
*Rosa26-Cre*$^{ERT}$;*Eng*$^{fl/fl}$ mice have been described previously[22,51] and
maintained in a C57BL/6 background. All procedures were approved
by the University of Cambridge Animal Welfare Ethical Review Board
under the authority of the United Kingdom Home Office Project
Licences PP7550697 and PP3923706. Mice were housed in a pathogen-
free barrier facility under a 12-h light and 12-h dark cycle at ambient
temperature and humidity with standard chow and water ad libitum.
Primary mouse lung microvascular ECs (MLECs) were isolated from the
*Rosa26-Cre*$^{ERT2}$;*Eng*$^{fl/fl}$ mice carrying the Immorto mouse transgene
using CD31 (553370, BD Biosciences)- and then ICAM2 (553326, BD
Biosciences)-coated beads following the protocols as describe
before[22]. MLECs were cultured at 33 °C in MV2 media (C22121, Pro-
moCell) with 10% FBS, 100 μg/ml heparin, 20 U/ml interferon-γ (315-05,
PeproTech) and 1 μM SB431542 (S4317-5MG, Sigma-Aldrich). *Eng*
knockout was induced by the addition of 2 μM 4-OH-tamoxifen
(H7904, Sigma-Aldrich) in the culture media for 72 h. Cells were see-
ded in the culture media without interferon-γ, SB431542 and 4-OH-
tamoxifen at 37 °C for 24 h followed by serum starvation overnight in
MV2 media containing 0.1% FBS and 100 μg/ml heparin at 37 °C before
signalling assays. Antibody information can be found in Supplemen-
tary Data 1.

## ENG displacement ELISA
A 96-well plate (655001, Greiner) was coated with anti-human CD105
antibody (555690, BD Biosciences, 0.17 μg per well) in PBS and incu-
bated at 4 °C overnight. The wells were then washed with PBS-T and
blocked with 1%BSA/PBS for 2 h at room temperature. sENG:BMP9 was
then added to the plate and incubated for 2 h at room temperature.
After washing with PBS-T, BMPRII or ActRIIB ECDs (at 0, 1, 5, 25 and
125×) was added to the wells for 30 min at room temperature. The plate

was washed and incubated with the biotinylated anti-BMP9 antibody
(BAF3209, Bio-Techne, 0.04 μg per well). After a 2-h incubation at
room temperature, the plate was washed and incubated with 1 mg/ml
ExtrAvidin-Alkaline phosphatase (E2636, Sigma-Aldrich) for 30 min.
The assay was developed with 4-Nitrophenyl phosphate (pNPP) dis-
odium salt hexahydrate (S0942, Sigma-Aldrich) in 1 M Diethanolamine,
0.5 mM MgCl$_2$ pH 9.8 and the absorbance was measured at 405 nm.

## ENG pull-down from primary ECs followed by mass spectro-
metry analysis
TRC105, a chimeric IgG1 monoclonal antibody that binds ENG, was a
kind gift from TRACON and was used to make the ENG affinity column
previously[22]. A 1 ml NHS-BSA column (NHS columns were purchased
from Cytiva, 17-0716-01) was generated using the same method as a
negative control. Three human primary ECs, HAOECs, HUVECs and
HPAECs were grown in T175 flasks in EGM2 with 10% FBS for two days.
Depending on the experiment, cells were maintained in EGM2 with 10%
FBS, or serum-starved in EGM2 with 0.1% FBS for 20 h, or serum-
starved in EGM2 with 0.1% FBS for 20 h followed by treatment with
0.25 ng/ml pro:BMP9 for 40 min. The cells were then harvested and
lysed in 1% Triton-X100/PBS with 2 mM EDTA and protease inhibitor
(11873580001, Merck). Lysed cells were centrifuged at 20,000 × g for
10 min at 4 °C. The supernatant from each experiment was divided into
two equal portions. One portion was passed through the 1 ml ENG
affinity column, while the other portion was passed through the 1 ml
NHS-BSA column under the identical condition. The columns were
washed with PBS, and bound proteins were eluted with 0.15 M glycine
pH 2.5. The eluates were collected, TCA precipitated and run on SDS-
PAGE under reducing condition for about 1 cm. The stained bands
were cut out and sent for LC-MS/MS analysis at the Proteomics Core
Facility, Department of Biochemistry, University of Cambridge. Total
of 18 samples were analysed across three different culturing condi-
tions, each condition containing $N = 3$ negative control and $N = 3$ bio-
logical repeats (three different EC lines in this case).

Sample preparations for mass spectrometry are as follows: SDS-
PAGE bands were cut into 1–2 mm pieces and washed in 50% acetoni-
trile/100 mM NH$_4$HCO$_3$ solution until clear. Reduction was performed
with 10 mM DTT in 100 mM NH$_4$HCO$_3$ at 37 °C for 1 h followed by
alkylation with 55 mM iodoacetamide in 100 mM NH$_4$HCO$_3$ at room
temperature for 45 min in the dark. Trypsin (V5111, Promega) was
added at 5 ng/ml 50 mM NH$_4$HCO$_3$, digestion was proceeded over-
night at 37 °C.

LC-MS/MS experiments were performed using a Dionex Ultimate
3000 RSLCnano UHPLC system coupled to an Orbitrap Fusion Lumos
mass spectrometer (Thermo Fisher Scientific). Peptides were first
loaded onto a PepMap 100 C18 pre-column (5 μm, 100 Å,
300 μm × 5 mm) with 0.1% formic acid for 3 min at 15 μl/min, then
separated on a PepMap C18 analytical column (2 μm, 100 Å,
75 μm × 50 cm) at 300 nl/min. Chromatographic separation used sol-
vent A (0.1% formic acid in water) and solvent B (80% acetonitrile, 0.1%

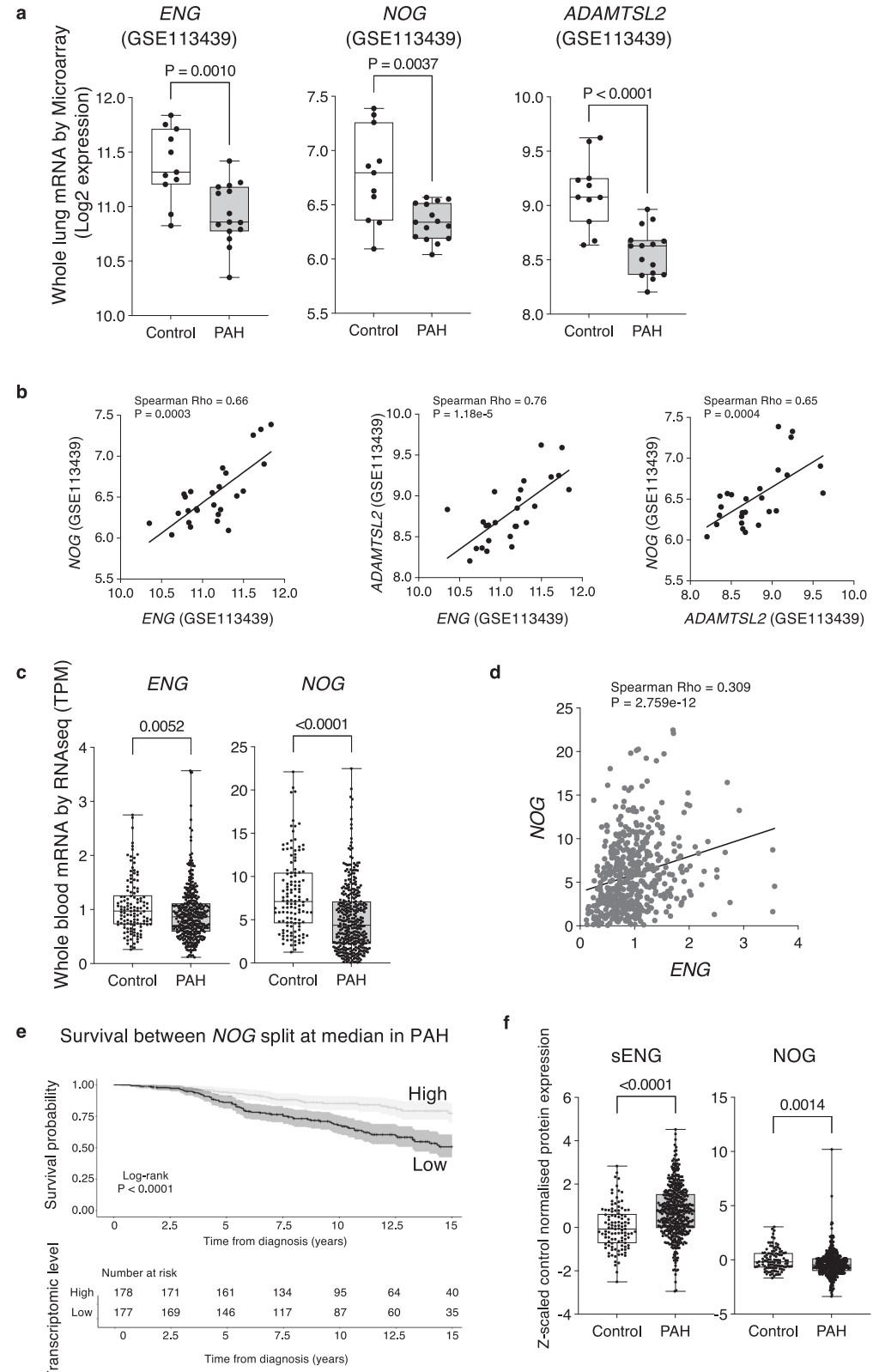

formic acid) with a 2–30% B gradient over 30 min. Eluting peptides were ionized using an Easy-Spray source.

MS1 scans were acquired in the Orbitrap at 120,000 resolution over m/z 380–1500. Data-dependent MS/MS (3s-cycle) used high energy collisional-induced dissociation (HCD) (Normalised Collision Energy (NCE) = 32%) with fragment detection at 15,000 resolution.

Singly charged ions, ions with >7 charges, and unassigned charge states were excluded; dynamic exclusion was set to 70 s.

Raw data were processed in Proteome Discoverer 2.5 (Thermo Fisher Scientific) and exported as mgf files for Mascot (v3.1, Matrix Science) searching against a contaminants database and the CCP_Homo_sapiens_20221011 database. Searches employed variable

**Fig. 9 | Analysis of Noggin and Adamtsl2 in healthy control and PAH patients' samples. a** Validating *ENG*, *NOG* and *ADAMTSL2* expression by analysing a published microarray dataset of lungs from $N = 15$ PAH patients and $N = 11$ healthy controls. **b** Spearman correlation tests (two-sided) of *ENG*, *NOG* and *ADAMTSL2*. Spearman Rho- and *p*-values are shown on the plots. **c** Examining *ENG* and *NOG* expression in published human whole blood transcriptomics from $N = 363$ PAH patients and $N = 126$ healthy controls. *ADAMTSL2* expression is too low to be reliably analysed. **d** Spearman correlation analysis (two-sided) of *ENG* and *NOG* expression in whole blood transcriptomics. Spearman Rho- and *p*-values are shown on the plot. **e** A Kaplan–Meier curve showing lower levels of *NOG* expression in whole blood mRNA is associated with poorer survival in PAH patients. Line defines probability of survival over time with 95% confidence intervals. **f** Examining NOG and sENG protein levels in a published human total blood proteomic dataset measured on SomaScan version 4, targeting 4152 unique human proteins[37,38]. Values are *Z*-value difference in PAH patients ($N = 463$) versus healthy controls ($N = 108$). ADAMTSL2 was not measured in this dataset. For (**a**, **c** and **f**), boxplots visualise the median, 25th and 75th quartile, and the whiskers represent the minimum and maximum values within 1.5 times the interquartile range; differences between groups were compared using a two-sided Mann–Whitney *U* test and normality was assessed using a Shapiro–Wilk test. Detailed description of statistical parameters of the boxplots is provided in Supplementary Table 1. Only significant *p*-values (<0.05) are shown.

modifications of oxidation (M) and deamidation (N/Q), fixed carbamidomethyl (C), and J-28 leucine/isoleucine ambiguity, allowing two missed cleavages. Mass tolerances were 20 ppm (precursors) and 0.1 Da (fragments). Protein abundances were calculated from peptide peak areas.

Peptide identifications were validated in Scaffold (v5.1.1) at ≥95% probability (Local FDR), and proteins at ≥99% probability with ≥2 peptides, using Protein Prophet[52]. The results were analysed and viewed using Scaffold 5 (Proteome Software.com).

### RNA extraction and RT-qPCR
Cells were lysed and RNA was extracted using RNeasy Mini Kit buffers (74104, Qiagen) following the manufacturer's protocol. 1 µg RNA was reverse transcribed using a High-Capacity cDNA Reverse Transcriptase kit (4368814, Thermo Fisher Scientific) in a Veriti thermal cycler (Applied Biosystems). Quantitative PCR was performed on 1/10 diluted cDNA samples using PowerUP SYBR Green Master Mix (A25777, Applied Biosystems) in a MicroAmp® Optical 384 Well Reaction plate (10005724, Applied Biosystems) on a Quant-Studio 6 Flex Real-Time PCR system (Applied Biosystems). The relative expression levels were calculated using the ΔCt method and fold changes in expression levels were calculated using the ΔΔCt method[53]. The sequences of the primers are in Supplementary Data 1. All the primers were synthesised by Sigma-Aldrich and were previously validated in published studies or in our assays.

### Signalling assay in MLECs
*Eng* knockout was induced by 2 µM 4-OH-tamoxifen with vehicle-treated cells as controls. Cells were seeded in 6-well plates and serum-starved overnight in MV2 media with 0.1% FBS and 100 µg/ml heparin at 37 °C. They were then treated with PBS, pro:BMP9 (0.2 ng/ml GFD concentration) or sENG(M):BMP9 (0.2 ng/ml GFD concentration, sENG in large excess to stabilise the complex) for 15 min before being harvested for analysis. The total protein was quantified using the DC protein assay (5000112, Bio-Rad). Equal amounts of lysate were fractionated on a 10% SDS-PAGE and Smad1/5 phosphorylation (9516S, Cell Signalling) was detected by immunoblotting. Knockout efficiency of ENG was confirmed by immunoblot using AF1320 (Bio-Techne) and RT-qPCR. β-actin (A5441, Sigma-Aldrich) was used as the loading control. Antibody information can be found in Supplementary Data 1.

### Immunofluorescence staining
MLECs were seeded in cells of a Milicell® EZ Slide (PEZGS0416, Merck) overnight. They were fixed the following day with acetone + methanol (1:1) for 5 min at room temperature. Following blocking with 1% BSA in PBS, the cells were stained with the corresponding primary antibodies (CD31, 553370, BD Bioscience; VE-Cadherin, 555289, BD Biosciences; ENG, 14-1051-82, Affymetrix eBioscience, Supplementary Data 1) for 2 h at room temperature. They were then washed and incubated with secondary antibodies (488, A-21208, Invitrogen; 568, A78946, Thermo Fisher Scientific) for 1 h at room temperature. Nuclei were counterstained with DAPI (40043-BT, Biotium) and images were acquired

using an Invitrogen EVOS M5000 Imaging System. Fluorescence images were acquired from cells isolated from three different mice. For each stained cell line, 4–5 images were taken from distinct regions for each protein. Microscope settings were kept constant for a given protein across all images and cells. Total integrated fluorescence intensity (IntDen) was measured using Fiji software (https://www.nature.com/articles/nmeth.2019). The IntDen was then normalised by the number of cells in each image to obtain the average intensity per cell (IntDen/cell). The final IntDen/cell value for each cell line represents the mean of all images collected for that line.

### RNA sequencing and bioinformatic analysis
MLECs from four independent isolations were used. For each MLEC line, *Eng* knockout was induced by 2 µM 4-OH-tamoxifen with vehicle-treated cells as controls. Cells were seeded in 6-well plates and serum-starved overnight in MV2 media with 0.1% FBS and 100 µg/ml heparin at 37 °C. They were then treated with PBS, pro:BMP9 or sENG(M):BMP9 (all at 0.2 ng/ml GFD concentration, sENG in large excess to stabilise the complex) for 1.5 h before being harvested and lysed. RNA extraction was performed as described above. The integrity and quality of the extracted RNA were assessed using an Agilent TapeStation 4200 System (agilent.com). Library preparation was conducted by Novogene (www.novogene.com) using the Novogene NGS Stranded RNA Library Prep Set (PT044). RNA sequencing was carried out by Novogene using an Illumina NovaSeq X Plus system to generate 150 bp paired end reads.

Raw data processing and quality control were performed by Novogen using in-house Perl scripts. The reads were aligned to the Mus musculus GRCm39 reference genome using Hisat2(v2.0.5). Mapped reads and sample metadata were imported into R (v.4.0.3) for downstream analysis. First, BiomaRt[54] (v.2.54.0) mapped the ensemble gene IDs to HGNC symbols. After performing quantile normalisation, COMBAT[55] from the sva package was applied to remove batch effects. Differential expression analysis was conducted using DESeq2[56] (v.1.26.0) comparing the different sample groups. The Benjamini-Hochberg method adjusted the raw *p*-values for the False Discovery Rate (FDR). Pathway analysis was assessed using String server (https://string-db.org/).

### Signalling in human primary ECs
HUVECs, HAOECs, HDMECs and HPAECs were plated in 6-well plates in EGM-2 media with 10% FBS. The following day cells were serum-starved with 0.1% FBS EGM-2 media for 20 h before being treated with PBS or pro:BMP9 at 0.1 or 1 ng/ml GFD concentration for the duration as indicated in the figure legends. Cells were then harvested for RNA extraction or immunoblotting.

### siRNA knockdown of receptors
HPAECs (in 60 mm plate) were transfected using DharmaFECT 3 transfection reagent according to DharmaFECT protocol with 50 nM (final concentration) siRNAs (ON-TARGET plus Smart pool human siRNAs from Dharmacon). The following day cells were re-seeded prior

to stimulation. Cells were serum-starved overnight before treatment with BMP9 GFD for 15 min and harvested in SDS-sample loading buffer or RNA extraction buffer. RNA was isolated using the NucleoSpin RNA II kit (BIOKE), followed by reverse transcription PCR (RevertAid First Strand cDNA Synthesis Kits, Fermentas). The cDNAs were analysed by qPCR, and calculations were performed using the CFX Manager software version 2.0 (Bio-Rad). All mRNA levels were analysed in triplicate and normalised to GAPDH expression.

### Bioinformatics analysis of human samples

*ENG*, *NOG*, and *ADAMTSL2* expression was analysed in a publicly available human lung microarray dataset (GSE113439[34] containing 15 PAH patients and 11 controls). Differences between normalised expression values were assessed using a two-sided Mann–Whitney *U* test. In a previously published whole blood transcriptomic dataset of PAH patients ($n = 363$) and healthy controls ($n = 126$)[35], differences in gene expression, as measured by transcripts per million (TPM), were assessed between cases and controls for *NOG* and *ENG* using a two-sided Mann–Whitney *U* test. In the lung microarray data and whole blood RNAseq, transcriptomic correlations between *ENG*, *NOG*, and *ADAMTSL2* were calculated using a two-sided Spearman test for both cases and controls using Log2 expression levels or TPM. Plasma proteomic levels of ENG and NOG were compared between patients ($n = 463$) and controls ($n = 108$) from a previously published dataset[37,38]. Differences in normalised protein level between cases and controls were assessed using a two-sided Mann–Whitney *U* test.

To assess whether *NOG* expression levels are associated with long-term outcomes, a survival analysis was performed. *NOG* expression levels were split in two groups on its median expression level in adult (diagnosed at ≥18 years) PAH patients with long-term follow-up data available ($n = 355$). Subsequently, a Kaplan–Meier curve was constructed comparing right-censored transplant-free survival from diagnosis up to 15 years post diagnosis using the Survival (v3.8-3) and Survminer (v0.5.0) packages in R/Rstudio (version 4.4.3). A log-rank test was performed. Subsequently, *NOG* expression levels were correlated with REVEAL 2 Lite scores using a Spearman test.

The clinical data were obtained from the National Cohort Study of Idiopathic and Heritable PAH (UK cohort; NCT01907295, East of England Ethics Committee: 13/EE/0203) with a group 1 PH diagnosis. The analyses in this manuscript included patients with IPAH and HPAH diagnosed at 18 years of age or older. Clinical data was collected prospectively at annual outpatient visits and the last study census took place on 01/07/2022. If patients were diagnosed before the study start date, diagnostic parameters were included retrospectively in the study.

### Statistical analysis

Quantitative data from qPCR and Immunoblot experiments were from at least three independent repeats as indicated in figure legend. The statistical analyses were performed using GraphPad Prism version 10.2.3. Two-sided unpaired *T*-test, one-way ANOVA or two-way ANOVA was used as indicated in the figure legends. $P < 0.05$ was considered statistically significant.

### Reporting summary

Further information on research design is available in the Nature Portfolio Reporting Summary linked to this article.

### Data availability

The data generated in this study are provided in the Supplementary Information and Source Data file. All antibodies and qPCR primers can be found in the Supplementary Data 1. The RNAseq dataset generated in this study has been deposited in the NCBI Gene Expression Omnibus database under accession code GSE289309. Links to data used in this study are as follows: 5HZW. 3KFD. 4YCG. 7POI. 4FAO. 7PPC. GSE289309. GSE113439. Sharing request for newly generated materials in this study can be sent to the corresponding author. Source data are provided with this paper.

### Code availability

All code used for the survival analyses and correlations between gene expression levels and REVEAL Lite 2 scores can be found at: https://github.com/EckartDeBie/Noggin_survival_in_PAH.

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

## Acknowledgements

The authors are grateful to Prof. Peter ten Dijke for critically reviewing the manuscript and providing many constructive comments, and to Prof. Nick Morrell, Mr Stephen Moore and Dr Bin Liu for help and support during re-establishing the mouse line locally. This work is supported by the British Heart Foundation (BHF) Senior Basic Science Research Fellowship to W.L. (FS/SBSRF/20/31005), a BHF PhD studentship (FS/4yPhD/F/20/34124B for K.K.), the BHF Cambridge Centre of Research Excellence award (RE/18/1/34212), and the US National Institutes of Health (NIH) grant K08HL169982 (to J.H.). I.K. and E.P. were supported by the European Molecular Biology Laboratory and the EMBL-EBI through core funding. E.BD is supported by the Gates Cambridge Trust - Bill & Melinda Gates Foundation grant OPP1144. C.J.R. is supported by a

BHF Basic Science Research Senior Fellowship (FS/SBSRF/22/31025). Cambridge National Institute for Health Research (NIHR) Biomedical Research Centre provided infrastructure support. The views expressed are those of the authors and not necessarily those of the NIHR or the Department of Health and Social Care. BioRender was used to generate the models and diagrams in the figures.

## Author contributions

J.G. collected most of the data and contributed to the writing of the manuscript. K.K. performed the experiments on human primary ECs. Z.T. contributed to the supplementary Fig. 5. I.K., A.V.-P. and E.P. analysed the RNAseq dataset. J.G., X.Y. and H.M.A. contributed to the generation of the mouse lung ECs. M.T., M.-J.G., E.G., and L.L. performed siRNA experiments. E.D.B., R.J.J., A.B, C.J.R., A.L., M.R.W., J.H. and M.R.T. analysed and collected human PAH datasets. W.L. conceived the idea, directed the study and wrote the paper. M.J.G., A.V.-P. and H.M.A. edited the manuscript.

## Competing interests

The authors declare no competing interests.
