## [Peer Review File · Nature Communications]

Endoglin as a BMP9 Co-Receptor in Vascular Endothelial Cells: Prodomain Displacement and TGFBR2 Recruitment

Corresponding Author: Dr Wei Li

Version 0:

Reviewer comments:

Reviewer #1

(Remarks to the Author)

This study provides compelling evidence for the dual role of endoglin (ENG) as a co-receptor for BMP9/10 signaling, bridging the gap between its previously reported functions in TGF- β and BMP pathways. The authors demonstrate that ENG efficiently displaces prodomains from pro-BMP9/10, recruits TGFBR2 to the signaling complex, and identifies Nog and Adamts12 as ENG-dependent BMP9 target genes. This manuscript presents a significant advance in understanding ENG's role in endothelial signaling. The work is well-executed and addresses a long-standing controversy in endothelial biology. However, some methodological details and interpretations require clarification or additional validation to strengthen the conclusions. Addressing the following points will elevate the impact and clarity of the work. I recommend acceptance pending revision.

1. Quantify prodomain displacement efficiency in Figure 1 using densitometric analysis of native PAGE bands, and perform kinetic assays to exclude potential prodomain reassociation post-displacement.
2. Provide structural evidence (e.g., SEC-MALS) to substantiate the hypothesis that ENG adopts multiple conformations in Figure 2a.
3. Label the analyte concentrations used in surface plasmon resonance (SPR) experiments in the legend of Figure 3.
4. Include statistical comparisons between the displacement efficiencies of BMPRII and ActRIIB in Figure 4b.
5. Validate the ENG-TGFBR2-ALK1 interaction in primary endothelial cells (ECs) by conducting co-immunoprecipitation (co-IP) assays with endogenous proteins under physiological conditions to support Figure 5e.
6. Optimize the schematic diagram of Figure 6d, and list the small diagrams separately on one side, clearly marking the important proteins such as the receptors.
7. Provide statistical analysis and comparison on the immunofluorescence intensity in Figure 7c.
8. Perform functional rescue experiments, such as ENG overexpression or TGFBR2 knockdown, to confirm the role of ENG in regulating Noggin and Adamts12 expression in Figure 8.
9. Explore the clinical relevance of Noggin and Adamts12 dysregulation by analyzing patient samples to enhance the clinical significance, and elucidate the mechanistic link between soluble ENG (sENG) levels and the pathophysiology of preeclampsia (PE) or pulmonary arterial hypertension (PAH).
10. Address spatial constraints of ENG's extracellular domain (ECD) in recruiting TGFBR2 and ALK1, and discuss potential ENG-independent BMP9 signaling pathways (e.g., via ActRIIA/BMPRII) in the Discussion.
11. Compare the ENG-TGFBR2-ALK1 axis with other TGF- β /BMP receptor complexes (e.g., BMPRII-ALK3) in the Discussion to provide broader mechanistic context.
12. Clarify in the Methods section whether crosslinker concentrations were titrated to optimize experimental conditions.

Reviewer #2

(Remarks to the Author)

In the study by Wei Li group, the authors elucidate a novel model of Endoglin (ENG) in the crosstalk of BMP9 and TGF- β signaling, as ENG firstly captures circulating BMP9 and BMP10 and efficiently displaces both their prodomains, secondly recruits TGFBR2 to BMP9 and 10 signalling by binding to TGFBR2 and ALK1 simultaneously. Their findings, in part, provide a long-standing question of how ENG regulates TGF- β superfamily signaling. They also identify the target genes of ENG-dependent BMP9 signaling, which provides insight for how to dissect context-dependent ALK1 signaling to understand pathogenesis of, for example, pulmonary arterial hypertension. While the manuscript is well-structured and the experiments

largely support the conclusions, several weaknesses should be addressed to strengthen the impact.

Major weaknesses:

1. The study utilizes Rosa26-CreERT2; Eng^{fl/fl}; immorto mice to generate Eng knockout MLECs for in vitro RNA-seq and signaling assays. The immorto system's potential interference raises concerns about whether MLECs used for this analysis truly reflect native endothelial behavior.
2. The co-enrichment of SMAD1, SMAD2 in the ENG pulldown complexes (Figure 5e) is insufficiently addressed.
3. The authors tested both BMP9 and BMP10 in some parts of the Results, however, they only tested BMP9 in other parts (Figure 4, 5 and 7). It would be helpful if the authors can explain the reason.
4. The current results provide a new framework to facilitate ENG research, however, which is still not clear whether this new mechanism/function of ENG will help for therapeutic applications. While the in vitro data are robust, the absence of in vivo validation limits translational impact.

Some minor issues in the text:

0. "Displacement of prodomain" may need some explanations for general readers, i.e., the full length of the BMP9/10 are cleaved between the prodomain and the growth factor (GF) domain, and these domains form a loose complex to stay inactive. A releasing mechanism of the prodomain is one of the layers of signaling control, which is indeed studied in this manuscript.
1. Line 39, 100, 217, 326, 399 and 471: the authors stated "BMP9 and 10", which is easy to cause ambiguity. Please revised to "BMP9 and BMP10" to keep consistence.
2. Line 75: The abbreviation "ICD" is introduced without its full term .
3. Figure 1d: The unlabeled bands in the sENG(M) + pro: BMP10 condition (resembling bands 7-9) should be clarified in the figure legend or text.
4. Figure 1g and 1h : Lane 11 (far right in each panel) is not annotated in the figure or legend.

Reviewer #3

(Remarks to the Author)

Version 1:

Reviewer comments:

Reviewer #1

(Remarks to the Author)

I have no other concerns.

Reviewer #2

(Remarks to the Author)

The authors substantially and successfully addressed the critiques raised by this reviewer.

Reviewer #3

(Remarks to the Author)

Point-by-point Responses to Reviewers' comments.

We thank the Editor and Reviewers for their thoughtful and constructive feedback. Below, we respond point-by-point and describe all changes to the text, figures, and Supplementary Information (Original comments from the reviewers are in black and our responses are in blue).

Reviewer #1 (Remarks to the Author):

This study provides compelling evidence for the dual role of endoglin (ENG) as a co-receptor for BMP9/10 signaling, bridging the gap between its previously reported functions in TGF- β and BMP pathways. The authors demonstrate that ENG efficiently displaces prodomains from pro-BMP9/10, recruits TGFBR11 to the signaling complex, and identifies Nog and Adamts12 as ENG-dependent BMP9 target genes. This manuscript presents a significant advance in understanding ENG's role in endothelial signaling. The work is well-executed and addresses a long-standing controversy in endothelial biology. However, some methodological details and interpretations require clarification or additional validation to strengthen the conclusions. Addressing the following points will elevate the impact and clarity of the work. I recommend acceptance pending revision.

1. Quantify prodomain displacement efficiency in Figure 1 using densitometric analysis of native PAGE bands, and perform kinetic assays to exclude potential prodomain reassociation post-displacement.

We now provide densitometric quantification from N = 3 independent experiments for the prodomain, sENG:BMP9 (or sENG:BMP10) (revised **Supplementary Figure 2c-e, 3b-d and 3g-h**).

Time-course assays up to 24 hour show no evidence of prodomain re-association for either proBMP9 or proBMP10 (revised **Supplementary Figure 2f-g, 3e-f**).

2. Provide structural evidence (e.g., SEC-MALS) to substantiate the hypothesis that ENG adopts multiple conformations in Figure 2a.

Firstly, we would like to clarify that Figure 2a is not a hypothesis that ENG adopts multiple conformations, but to illustrate the theoretical "possible forms of complexes" (as marked on the figure) from binding stoichiometry point of view to guide the analysis of the data in the rest of Figure 2. As shown in Figure 2 results, our experimental data can exclude the forms IV and VI. We don't have enough resolution to distinguish between forms I, II and III.

To acknowledge this and to avoid potential confusion (**thank you for pointing this out**), we added the following sentence at the end of this paragraph. "Further experiments are required to ascertain whether forms I, II and III are all possible and whether one of them is a preferred form."(Page 8, line 212)

It is worth noting here that SEC-MALS of BMP9:ENG complex has been performed in Saito et al, Cell Report, 2017, PMID: 28564608. This paper from Prof. Luca Jovine's group reported crystal structures of ENG-OR in complex with BMP9 as well as ENG ZP domain on its own. In Figure 4E from this paper, they detected sENG-ECD-BMP9 complex showing two peaks in SEC-MALS analysis. In addition, the ZP domain alone secreted as both monomer and dimer, although monomer preferentially crystallised. In this report, the authors also suggest that "*Elucidation of*

the details of the signal transduction mechanism will require further structural studies of the intact ternary complex. This may adopt a range of conformations not recapitulated by our crystal structure-based model as a result of flexibility at the ENG OR/ZP junction—possibly underlying the two ECTO-BMP9 states observed by SEC-MALS (Figure 4E)—as well as membrane anchoring.” (page 1192 of the paper)

3. Label the analyte concentrations used in surface plasmon resonance (SPR) experiments in the legend of Figure 3.

Thank you. We have now added the analyte concentrations in the figure legend.

4. Include statistical comparisons between the displacement efficiencies of BMPRII and ActRIIB in Figure 4b.

Statistical comparisons at matched molar excesses are now included in Figure 4b and legend. No significant differences in displacement efficiency were detected, with a small ELISA signal enhancement at 1× ActRIIB for sENG(M):BMP9 noted in the legend and text.

5. Validate the ENG-TGFBR11-ALK1 interaction in primary endothelial cells (ECs) by conducting co-immunoprecipitation (co-IP) assays with endogenous proteins under physiological conditions to support Figure 5e.

Thank you for your suggestion. To mimic the physiological condition, we performed further co-IP with cells cultured under (i) 10% FBS standard endothelial cell growth medium (EGM-2) and (ii) 0.1% FBS EGM-2 medium with 0.25 ng/ml BMP9 (within healthy human plasma range), in three human primary endothelial cells as before.

This new data is now included in the **revised Figure 5e&f**, and we have added further western blots to report the Smad1/5 phosphorylation status of these cells (**Figure 5g**) as well as diagrams to summarise the pull-down results (**Figure 5h**). We consistently recovered endogenous TGFBR11 with ENG across conditions, whereas ALK1 was not detected under 10% FBS or BMP9 treatment (**revised Figure 5e–h**; Smad1/5 phosphorylation controls in Figure 5g). We discuss that constitutive/ligand-induced ALK1 activation likely alters detectability (phosphorylated ALK1 may dissociate or be poorly recovered), and we note that our pulldown antibody (TRC105) competes with the ENG–BMP9 interface, precluding recovery of extracellular ENG:BMP9:ALK1 complexes in this assay. The model in **Figure 6f** has been adjusted to avoid implying an intracellular ALK1–TGFBR11 interaction in the presence of BMP9. Overall, the data support the notion that ENG recruits TGFBR11 into the BMP9–ALK1 signalling complex.

6. Optimize the schematic diagram of Figure 6d, and list the small diagrams separately on one side, clearly marking the important proteins such as the receptors.

Mini-diagrams have been added adjacent to the relevant data (**Figure 1i**; **Figure 3a–c**; **Figure 5h**; **Figure 6b, e**) and the final model (**Figure 6f**) was updated for consistency.

Proteins/receptors are clearly labelled throughout.

7. Provide statistical analysis and comparison on the immunofluorescence intensity in Figure 7c.

The quantification and statistical analysis are now added in **Supplementary Figure 8a**, from staining of N=3 batches of cells isolated from 3 difference mice.

The quantification method is included in the Immunofluorescence staining section, also explained in the legend. Manuscript text has been modified accordingly.

8. Perform functional rescue experiments, such as ENG overexpression or TGFBR2 knockdown, to confirm the role of ENG in regulating Noggin and Adamtsl2 expression in Figure 8.

1) We have performed siTGFBR2 in PAECs and did not observe any difference in *NOG* or *ADAMTSL2* induction by BMP9. We think the effect of ENG loss may not be fully captured by siTGFBR2. This is because the effects from ENG loss can affect BMP9 signalling from at least two aspects: 1) less efficient capturing of BMP9 therefore less signalling strength (e.g. reduced Smad1/5 phosphorylation); 2) mediating TGFBR2 to phosphorylate ALK1. siTGFBR2 can only test the point 2, it will not capture point 1.

2) We tried to transfect the human ENG-containing plasmid into the MLECs after Eng knockout, using Lipofectamine LTX Reagent with PLUS Reagent (ThermoFisher 15338100) as well as Neon NxT electroporation System with the transfection kit (ThermoFisher, N10025). These cells were very susceptible to Lipofectamine LTX, and over 90% of the cells died after transfection. We attempted various protocols and varying amounts of ENG and control vector plasmids. Approximately two-thirds of the cells died after electroporation in both the ENG- and the control vector-transfected cells across all tested conditions. Although we can detect the successful transfection of human ENG, these remaining cells were not healthy, as reflected by lower overall protein contents and lower RNA yield. When we treated remaining cells with BMP9, we found that transfecting the vector control alone affected the basal expression of both *Nog* and *Adamtsl2*. Therefore, it is not possible to do the rescue experiment using this approach.

3) Although the gene manipulation approaches in the primary endothelial cells proved to be challenging, we have managed to analyse some human datasets, especially microarray data from human lung tissues (both healthy controls and PAH patients). As lung tissues have large areas of blood vessels, especially small vessels and capillaries, they contain a large proportion of endothelial cells. The lung tissue microarray data would contain signals from endothelial cells without any further effects introduced by *in vitro* culturing, hence providing an even better tool to validate our findings. If our model is correct, we should find the mRNA expression of *NOG* and *ADAMTSL2* to be positively correlated with the expression of *ENG*. Indeed, when we analysed the published microarray data GSE113439 which contains all three genes, we found strong positive correlation for both genes with ENG, using two different correlation analyses methods (Pearson $r=0.71$ to 0.74 for *ADAMTSL2* and *NOG* respectively, and $p<0.0001$ for both cases. If using Spearman test, Rho values for *NOG* and *ADAMTSL2* are 0.66 and 0.76 , respectively). More interestingly, we found that the mRNA levels for *ENG*, *NOG* and *ADAMTSL2* are all reduced in PAH patient lungs. These novel exciting data, along with further analysis of PAH blood samples (more information on this in the next point), are now included in the **new Figure 9**.

In summary, although we could not add further rescue experiment data to the revised manuscript, we believe these new human data provided more substantial evidence than *in vitro* functional rescue experiments to further support our findings.

9. Explore the clinical relevance of Noggin and Adamtsl2 dysregulation by analyzing patient samples to enhance the clinical significance, and elucidate the mechanistic link between

soluble ENG (sENG) levels and the pathophysiology of preeclampsia (PE) or pulmonary arterial hypertension (PAH).

Thank you for the suggestion. We have now analysed the expression of Noggin and Adamtsl2 as well as ENG in PAH patients' samples by: 1) analysing a published microarray dataset which compared lung mRNA between 15 PAH patients and 11 healthy controls; 2) analysing the total blood transcriptomics and proteomics, comparing PAH patients versus healthy controls. A new figure (**Figure 9**) is added in the revision with accompanying texts in the results and discussion.

We show that in the GSE113439 microarray dataset, mRNA levels for *ENG*, *NOG* and *ADAMTSL2* are all significantly lower in PAH patients' lung tissues compared with healthy controls (Figure 9a). As mentioned above, there are strong positive correlations between the expression of *NOG* and *ADAMTSL2* with the expression of *ENG*, consistent with our findings.

In the whole blood RNAseq data from 363 PAH samples versus 126 control samples, the expression levels for both *ENG* and *NOG* are significantly lower in PAH patients (Figure 9c), and the mRNA expression levels of *ENG* and *NOG* are also significantly correlated. *ADAMTSL2* expression level was too low to be reliably analysed. We also found that lower levels of *NOG* mRNA are associated with poorer survival in PAH patients (**Figure 9e**). Proteomic data from a published dataset containing 463 PAH patients and 108 healthy controls revealed a significant decrease in the levels of Noggin protein (Figure 9f). Our data suggest that plasma *NOG* could be an important biomarker for PAH, and this may be linked to the *ENG*/*BMP9* signalling axis, which was known to be dysregulated from the human genetic studies.

10. Address spatial constraints of *ENG*'s extracellular domain (ECD) in recruiting *TGFBR2* and *ALK1*, and discuss potential *ENG*-independent *BMP9* signaling pathways (e.g., via *ActR2A/BMPRII*) in the Discussion.

In the discussion, to address the spatial constraints of *ENG* ECD in recruiting *TGFBR2* and *ALK1*, we added the following sentence (page 17, line 1) "Structural analysis shows that *ENG*, *TGFBR2* and *ALK1* ECDs all bind to ligands at different sites and there will not be any structural constraint extracellularly for *ENG* to bring *TGFBR2* and *ALK1* together." This is also illustrated in new **Supplementary Figure 10**.

To address potential *ENG*-independent *BMP9* signalling, the following sentence was added (page 17, line 477): "Interestingly, *BMP9* signalling by reduced by around 50% in *Eng*^{-/-} MLECs, suggesting that both *ENG*-dependent and *ENG*-independent pathways play significant roles in *BMP9* signalling."

11. Compare the *ENG*-*TGFBR2*-*ALK1* axis with other *TGF-β*/*BMP* receptor complexes (e.g., *BMPRII*-*ALK3*) in the Discussion to provide broader mechanistic context.

We have added a paragraph in the discussion (page 18, line 515) to compare the *ENG*-*TGFBR2*-*ALK1* axis with other *TGF-β*/*BMP* receptor complexes in non-endothelial context, highlighting common principles and unique features and putting our work in a broader mechanistic context.

12. Clarify in the Methods section whether crosslinker concentrations were titrated to optimize experimental conditions.

Thank you for the suggestions. Methods now state that concentrations were titrated.

Related to Figure 6a, we agree that titrating the concentrations of the crosslinker is important to show that the incomplete crosslinking of sENG(D) is not due to the sub-optimal crosslinker concentrations. We now show that all crosslinkers at all tested concentrations can crosslink no more than 50% of sENG(D), consistent with our initial findings. We now added this new data in the **Supplementary data 7b** and cite it along with Figure 6a.

Reviewer #2 (Remarks to the Author):

In the study by Wei Li group, the authors elucidate a novel model of Endoglin (ENG) in the crosstalk of BMP9 and TGF- β signaling, as ENG firstly captures circulating BMP9 and BMP10 and efficiently displaces both their prodomains, secondly recruits TGFBR2 to BMP9 and 10 signalling by binding to TGFBR2 and ALK1 simultaneously. Their findings, in part, provide a long-standing question of how ENG regulates TBG-beta superfamily signaling. They also identify the target genes of ENG-dependent BMP9 signaling, which provides insight for how to dissect context-dependent ALK1 signaling to understand pathogenesis of, for example, pulmonary arterial hypertension. While the manuscript is well-structured and the experiments largely support the conclusions, several weaknesses should be addressed to strengthen the impact.

Major weaknesses:

1. The study utilizes Rosa26-CreERT2; *Eng*^{fl/fl}; immorto mice to generate *Eng* knockout MLECs for in vitro RNA-seq and signaling assays. The immorto system's potential interference raises concerns about whether MLECs used for this analysis truly reflect native endothelial behavior.

We clarify that the Immorto system is used strictly to enable discovery while minimising animal usage and passage-related drift; all functional assays are at 37 °C, where the ts T-antigen is inactive, and *Eng* deletion is induced immediately before experiments. Both control and KO carry the same ts allele, so differential phenotypes are attributable to ENG. MLECs are a powerful tool to facilitate the discovery of novel mechanisms which can then be further validated using other more physiologically relevant cells.

In our case, using MLECs we identified *Nog* and *Adamtsl2* to be preferentially dependent on *Eng*. We recognise the importance of validating such data using human primary endothelial cells which is what we did in Figure 8g-h. We indeed found that in HDMECs, which express lowest levels of *ENG*, both *NOG* and *ADAMTSL2* are minimally induced by BMP9, despite *ID1* being induced at similar levels.

In the revision, we performed further validation using human lung tissue microarray data (GSE113439). Lung tissue has a significant proportion of endothelial cells; and these endothelial cells would all be under physiological or pathological conditions (such as PAH) without any ex vivo culturing. Because BMP9 circulate at active concentrations, BMP9 signalling should be constitutively active in the lung ECs.

If our finding is correct that *NOG* and *ADAMTSL2* expression is *ENG*-dependent, we would expect the mRNA expression levels of *NOG* and *ADAMTSL2* to correlate with the mRNA levels of *ENG*. Indeed, we found strong positive correlation of *ADAMTSL2* and *NOG* with the expression

of *ENG*, with Pearson $r = 0.71$ to 0.74 for *ADAMTSL2* and *NOG*, respectively, both with P values less than 0.0001. Similarly using the Spearman test (which takes outlier into account), Rho values for *NOG* and *ADAMTSL2* are 0.66 and 0.76, respectively, with P values of 0.0003 and 0.00001 respectively. Such human data provide strong support to validate our findings from the MLECs. Of note, only Spearman test result was presented in the revised Figure 9.

2. The co-enrichment of SMAD1, SMAD2 in the *ENG* pulldown complexes (Figure 5e) is insufficiently addressed.

This is an interesting point. Of note, this was observed in the medium containing 0.1% FBS. As mentioned above in response to Reviewer 1, we have performed further pull-down/mass spec experiments for cells harvested in more “physiological-like” conditions, such as 10% FBS and 0.25 ng/ml pro:BMP9 in 0.1% FBS, where BMP9 signalling is constitutively activated.

As shown in the revised Figure 5e, under 10% FBS, we did not detect any SMADS in any of the three cell types, whereas under BMP9+0.1% FBS condition, we detected SMAD1 and SMAD2 in HAOECs, SMAD1 and SMAD5 in HPAECs and none of the SMADS in HUVECs.

We have provided the information on the peptides detected in Supplementary Figure 6.

R-SMADs are predicted to interact with the type I receptor under non-phosphorylated states (when signalling is not active) because after phosphorylation by the type I receptors, they will dissociate from the type I receptor and go into the nucleus to regulate transcription.

Mixed R-SMADs complex has been described before (PMID: 29376829), but its function in the physiological responses is still not fully understood. Therefore, we cannot exclude the possibility that there is direct interaction between SMAD1 and SMAD2.

We have added a paragraph in the discussion to address the different observations on ALK1, SMAD1, SMAD2 and SMAD5 from the IP/mass spec experiment while avoiding mechanistic over-interpretation (Page 17, line 489).

3. The authors tested both BMP9 and BMP10 in some parts of the Results, however, they only tested BMP9 in other parts (Figure 4, 5 and 7). It would be helpful if the authors can explain the reason.

Text (page 9, line 245) now explains that biochemical differences in prodomain displacement (BMPRII vs pro:BMP9 and Pro:BMP10) and s*ENG*:ligand binding to ALK1-Fc motivated focusing mechanistic depth on BMP9. Because BMP10 signalling regulation differs from BMP9, role of *ENG* in BMP10 will require separate in-depth investigation.

4. The current results provide a new framework to facilitate *ENG* research, however, which is still not clear whether this new mechanism/function of *ENG* will help for therapeutic applications. While the in vitro data are robust, the absence of in vivo validation limits translational impact.

Thank you for kindly pointing out that “The current results provide a new framework to facilitate *ENG* research”.

To increase the translational impact, we have included additional data from analysing human PAH patients’ data in the revision (Figure 9) to enhance the clinical relevance and translational impact. We show that *NOG* and *ADAMTSL2* expression in human lungs has strong positive

correlations to ENG expression, and the expression of all three genes is reduced in PAH patients. We also show that in whole blood transcriptomics, NOG and ENG expression are significantly correlated and reduced in PAH patients. Importantly, we identified that plasma protein levels of NOG are reduced in PAH. Together, our data strongly suggest plasma NOG and ENG gene and protein levels could be helpful PAH biomarkers that are linked to the TGF- β / BMP pathway, the dysregulated pathway in PAH identified by human genetics.

Some minor issues in the text:

0. “Displacement of prodomain” may need some explanations for general readers, i.e., the full length of the BMP9/10 are cleaved between the prodomain and the growth factor (GF) domain, and these domains form a loose complex to stay inactive. A releasing mechanism of the prodomain is one of the layers of signaling control, which is indeed studied in this manuscript. Thank you for this excellent suggestion. We added a schematic of BMP9/10 processing and clarified “prodomain displacement” for general readers, and explained specifically that the prodomain needs to be displaced because the unprocessed proBMP9 and proBMP10 are not active with two reference papers cited (**new Supp. Figure 1a**; page 4, line 91-96)

1. Line 39, 100, 217, 326, 399 and 471: the authors stated “BMP9 and 10”, which is easy to cause ambiguity. Please revised to “BMP9 and BMP10” to keep consistence.

Thank you. We have revised all these to “BMP9 and BMP10” as suggested.

2. Line 75: The abbreviation “ICD” is introduced without its full term .

Apologies for the oversight. “Intracellular domain (ICD)” is now introduced on first use.

3. Figure 1d: The unlabeled bands in the sENG(M) + pro: BMP10 condition (resembling bands 7-9) should be clarified in the figure legend or text.

We have clarified the unlabelled bands in the figure/figure legends (that they are sENG(D) and its complex with BMP9 or BMP10). We have confirmed this by performing an additional test with a purer version of sENG(M) containing less sENG(D) (see below, the additional bands are much fainter).

4. Figure 1g and 1h: Lane 11 (far right in each panel) is not annotated in the figure or legend.

Apologies for the oversight. These are type II receptor-only controls without BMP9 or BMP10. Both panels are now clearly annotated.

Reviewer #3 (Remarks to the Author):

I co-reviewed this manuscript with one of the reviewers who provided the listed reports. This is

part of the Nature Communications initiative to facilitate training in peer review and to provide appropriate recognition for Early Career Researchers who co-review manuscripts.

Thank you for reviewing our manuscript and providing constructive comments.